# Versatile Learned Video Compression

## Abstract

Learned video compression methods have demonstrated great promise in catching up with traditional video codecs in their rate-distortion (R-D) performance. However, existing learned video compression schemes are limited by the binding of the prediction mode and the fixed network framework. They are unable to support various inter prediction modes and thus inapplicable for various scenarios. In this paper, to break this limitation, we propose a versatile learned video compression (VLVC) framework that uses one model to support all possible prediction modes. Specifically, to realize versatile compression, we first build a motion compensation module that applies multiple 3D motion vector fields (*i.e.*, voxel flows) for weighted trilinear warping in spatial-temporal space. The voxel flows convey the information of temporal reference position that helps to decouple inter prediction modes away from framework designing. Secondly, in case of multiple-reference-frame prediction, we apply a flow prediction module to predict accurate motion trajectories with a unified polynomial function. We show that the flow prediction module can largely reduce the transmission cost of voxel flows. Experimental results demonstrate that our proposed VLVC not only supports versatile compression in various settings but also achieves comparable R-D performance with the latest Versatile Video Coding (VVC) standard in terms of MS-SSIM.

## 1 Introduction

Video occupies more than 80% of network traffic and the amount of video data is increasing rapidly [1]. Thus, the storage and transmission of video become more challenging. A series of hybrid video coding standards have been proposed, such as AVC/H.264 [2], HEVC/H.265 [3] and the latest video coding standard VVC/H.266 [4]. These traditional standards are manually designed, evolving for decades. However, the development within the hybrid coding framework is gradually saturated. Recently, the performance of video compression is mainly improved by designing more complex prediction modes, leading to increased coding complexity.

Deep neural networks are currently promoting the development of data compression. Despite the remarkable progress on the field of learned image compression [5–10], the area of learned video compression is still in early stages. Existing methods for learned video compression can be grouped into three categories, including frame interpolation-based methods [11, 12], 3D autoencoder-based methods [13, 14], and predictive coding methods with optical flows [15–17]. So far, among them, video compression with optical flow presents the best performance [18], where optical flow represents pixel-wise motion vector (MV) fields utilized for inter frame prediction. In this paper, we also focus on this predictive coding architecture. Previous works with optical flow are proposed to support specific prediction mode, including unidirectional or bidirectional, single or multiple frame prediction. They are too cumbersome to support versatile compression in various settings since they bind the inter prediction mode with the fixed network framework. It is important to design a more flexible model to handle all possible settings like traditional codecs. In this paper, we propose a versatile learned video

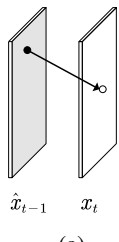 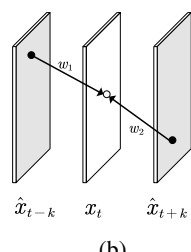 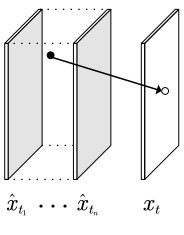 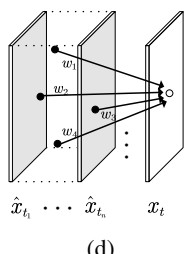

$\hat{x}_{t-1}$ $x_t$     $\hat{x}_{t-k}$ $x_t$ $\hat{x}_{t+k}$     $\hat{x}_{t_1}\cdots\hat{x}_{t_n}$ $x_t$     $\hat{x}_{t_1}\cdots\hat{x}_{t_n}$ $x_t$

(a)    (b)    (c)    (d)

Figure 1: Different motion compensation (inter frame prediction) methods. (a) Unidirectional prediction with 2D optical flow [15]. (b) Bidirectional prediction with two optical flows and weight coefficients [12]. (c) Prediction with a single voxel flow, freely sampling the reference frames in space-time. (d) Prediction with multiple voxel flows via weighted trilinear warping.

compression (VLVC) framework that achieves coding flexibility as well as compression performance. A voxel flow based motion compensation module is adopted for higher flexibility, which is then extended into multiple voxel flows to perform weighted trilinear warping. In addition, in case of multiple-reference-frame prediction, a polynomial motion trajectories based flow prediction module is designed for better compression performance. Our motivations are described as follows.

**Motion compensation with multiple voxel flows.** Previous works such as [15] apply 2D optical flows for low-delay prediction using single reference frame (unidirectional prediction, see Fig. 1a). For the practical random access scenario, bidirectional reference frames are available for more accurate frame interpolation [12] (Fig. 1b). However, the reference positions in these works are determined by pre-defined prediction modes. They cannot adapt to various inter prediction modes where reference positions are various. In this paper, we apply 3D voxel flows to describe not only the spatial MVs, but also the information of temporal reference positions (Fig. 1c). We perform voxel flow based motion compensation via trilinear warping, which is applicable to single or multiple, unidirectional or bidirectional reference frames. Unlike [16] that adopts scale space flow with trilinear warping, we apply voxel flows for inter prediction in spatial-temporal space, which naturally renders our model more robust to different coding scenarios. Furthermore, beyond using single MV in every position of the current frame, we propose to use multiple voxel flows to describe multiple possible reference relationships (Fig. 1d). Then the target pixel is synthesized by weighted fusing of the warping results. We show that without increasing the coding cost of motion information, the motion compensation is thus more accurate, yielding less residuals and more efficient compression.

**Flow prediction with polynomial motion trajectories.** Exploiting multiple reference frames usually achieves better compression performance since more reference information is provided. A versatile learned video compression model should cover this multi-reference case. While previous work [17] designs a complex flow prediction network to reduce the redundancies of 2D MV fields, the number and structure of reference frames are inherent and fixed within the framework. In this paper, we design a more intelligent method for flow prediction, *i.e.*, modelling the prediction modes with polynomial coefficients. We formulate different motion trajectories in a time interval by a unified polynomial function. The polynomial coefficients are solved by establishing a multivariate equation (see Section 3.2). Since this polynomial function models the accurate motion trajectories, it serves as a basic discipline that constrains the predicted motion to be reasonable. We show the transmission cost of voxel flows is reduced obviously with the help of additional motion trajectory information.

Thanks to the above two technical contributions, our proposed VLVC is not only applicable for various practical compression scenarios with different inter prediction modes, but also delivers impressive R-D performance on standard test sequences. Extensive experimental results demonstrate that our method is the first one to achieve comparable performance with VVC in terms of MS-SSIM in both low delay and random access configurations. Comprehensive ablation studies and discussions are provided to verify the effectiveness of our method.

## 2 Related Work

**Learned Image Compression**   Recent advances in learned image compression [5–7], have shown the great success of nonlinear transform coding. Many existing methods are built upon hyperprior-

based coding framework [6], which are improved with more efficient entropy models [7, 8], variable-rate compression [19] and more effective quantization [9, 10]. While the widely used autoregressive entropy models provide significant performance gain in image coding, the high decoding complexity is not suitable for practical video compression. We thus employ the hyperprior model [6] without context models in our video compression framework.

**Learned Video Compression** Existing approaches [11–15, 17, 18, 20–24] can be roughly divided into three categories: frame interpolation-based methods [11, 12, 24], 3D autoencoder-based methods [13, 14], and predictive coding methods with optical flows [15–17]. Currently, researchers are more interested in the latter two methods. Although 3D autoencoder-based methods requires less time complexity, they barely achieve comparable performance with x265 in MS-SSIM [14]. Meanwhile, predictive coding methods with optical flows have outperformed HM in terms of PSNR [18].

Predictive-based video compression approaches [15, 20–24, 11, 12, 17] sequentially perform motion estimation, motion compression, motion compensation and residual compression. Chen et al. [24] first propose to predict block of pixels using learned neural network (DNN), and the residual is compressed by a RNN-based autoencoder. Wu et al. [11] propose a interpolation-based approach using traditional MVs. Lu et al. [15] propose an fully end-to-end trainable framework, where all key components in the classical video codec are implemented with neural networks. Rippel et al. [21] jointly compress the motion and residual information, and propose a latent state to memorize the information from the past. Djelouah et al. [12] perform interpolation by the decoded optical flow and blending coefficients. They reuse the same autoencoder of I-frame compression and directly quantize the corresponding latent space residual. Liu et al. [23] combine the optical flow estimation and motion compression networks into one-stage, and remove the redundancy of quantized flow representations using joint spatial-temporal priors. Yang et al. [20] propose a video compression framework with three hierarchical quality layers and recurrent enhancement. In [17], multiple frames motion prediction are introduced into the P-frame coding. Lu et al. [22] propose an content adaptive and error propagation aware method to reduce error accumulation and achieve adaptive coding. Agustsson et al. [16] replace the bilinear warping operation with scale-space flow which allows the model adaptively blur the reference content for better warping results. However, most existing methods are designed for particular prediction modes, resulting in inflexibility for different scenarios.

**Video Interpolation** The task of video interpolation is closely related to video compression. One pioneering work [25] proposes to use deep voxel flow to synthesize new video frames. Some works of video interpolation [26–28] directly generate the spatially-adaptive convolutional kernels for each motion vectors by neural networks. Most recently, [29, 30] proposed to relax the kernel shape, allowing the models to freely select multiple sampling points in space or space-time. In this paper, our employed multiple voxel flows is motivated by the accurate interpolation result in [30].

## 3 Versatile Learned Video Compression

To compress video, the original video sequence is first divided into groups of pictures (gop). Let $\boldsymbol{x} = \{\boldsymbol{x}_1, \boldsymbol{x}_2, ..., \boldsymbol{x}_T\}$ denote the frames in one gop unit where the gop size is $T$. To take advantage of previous decoded frames, our model predicts the current frame $\boldsymbol{x}_t$ from $n$ reference frame(s), *i.e.*, the lossy reconstruction results compared to the original frames. Here, we denote the reference frames as $\{\hat{\boldsymbol{x}}_{t_1}, \hat{\boldsymbol{x}}_{t_2}, ..., \hat{\boldsymbol{x}}_{t_n}\}$, where $\{t_1, t_2, ..., t_n\}$ is the index of temporal reference position. If multiple frames are taken as the reference (*i.e.*, $n > 1$), the reference frames are divided into two groups: one is used only for flow prediction, and the other is used for both flow prediction and motion compensation. In other words, the reference involved for motion compensation is only a sub-set of $\{\hat{\boldsymbol{x}}_{t_1}, \hat{\boldsymbol{x}}_{t_2}, ..., \hat{\boldsymbol{x}}_{t_n}\}$, which could be concatenated into a volume denoted by $\hat{\boldsymbol{X}}_t$. If only one reference frame is available, the volume for warping $\hat{\boldsymbol{X}}_t = \{\hat{\boldsymbol{x}}_{t-1}\}$.

An overview of our video compression framework is shown in Fig. 2. Previous work [16] demonstrates that an implicit flow encoder can outperform a pre-trained optical flow network and simplify the network structures simultaneously. In our paper, we also abandon the use of a pre-trained optical flow network in motion encoder. The motion encoder and decoder are similar to image compression network [5]. While the work of [16] sends current frame and previous reconstruction into motion encoder, we make some modifications on the input of motion encoder. Specifically, in our framework, the motion encoder is fed with the current frame $\boldsymbol{x}_t$ concatenated with predicted frames (represented

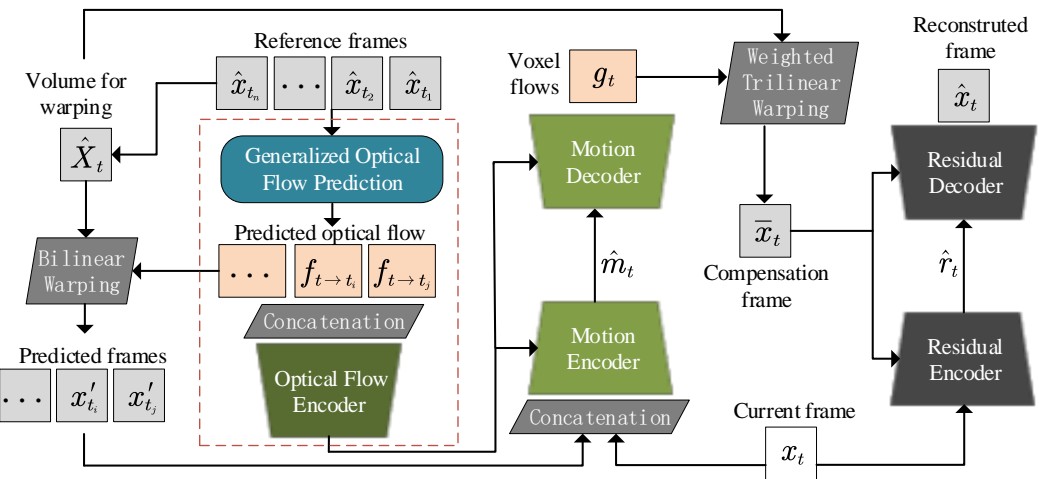

Figure 2: Overview of our inter-frame coding framework.

as $\boldsymbol{x}_t'$ in Fig. 2). Here, the predicted frame $\boldsymbol{x}_t'$ is an estimation of current frame $\boldsymbol{x}_t$. The flow prediction module will predict 2D optical flows $f_{t \to t_i}$ to warp corresponding reference frames $\hat{x}_{t_i}$, each of which will generate a predicted frame $\boldsymbol{x}_{t_i}'$. The predicted frames reveal how much information the decoder knows about current frame.

On the decoder side, a motion decoder will generate voxel flows which is used for motion compensation via trilinear warping (details are explained in Section 3.1). In addition, when multiple reference frames are available, the flow prediction module is turned on (the red dashed box). As shown in Fig. 2, the flow prediction module is auxiliary for motion encoding and decoding, which reduces the transmission cost of the quantized motion latent $\hat{\boldsymbol{m}}_t$. The specific mechanism of flow prediction can be found in Section 3.2. After motion compensation, we obtain a prediction of current frame as $\bar{\boldsymbol{x}}_t$. The residual encoder and decoder are then used to compress the remaining residuals between the original frame $\boldsymbol{x}_t$ and the predicted frame $\bar{\boldsymbol{x}}_t$, yielding the final reconstructed frame $\hat{\boldsymbol{x}}_t$ and quantized latent $\hat{\boldsymbol{r}}_t$.

## 3.1 Motion compensation with multiple voxel flows

Voxel flow [25] is a per-pixel 3-D motion vector that describes relationships in spatial-temporal domain. Compared to 2D optical flow, voxel flow can inherently allow the codec to be aware of the sampling positions in the temporal dimension for various prediction modes. Given arbitrary number of reference frames, the model is expected to select the optimal reference frame for better reconstructing the current frame to be compressed. Such a 3-D motion descriptor helps to build a prediction-model-agnostic video compression framework, *i.e.*, versatile learned video codec.

In addition, single flow field is hard to represent complex motion (e.g. blurry motion), which may result in inaccurate motion compensation or high coding cost of motions. When reconstructing a local region, its reference information may not come from only one source. Considering a practical scene where multiple objects of the same types appear at the same time, more than one areas could be referred for reconstructing the local region. Thereby, in this work, we further propose to use multiple voxel flows to perform weighted trilinear warping by sampling in $\boldsymbol{X}_t$ for multiple times. We remind our readers that $\boldsymbol{X}_t$ is a volume consisting of some reference frames. Assume the dimension of $\boldsymbol{X}_t$ is $D \times H \times W$ (usually reshaped into $H \times W \times D$ for warping), where $D$ is the number of reference frames used for motion compensation. the motion decoder will generate multiple voxel flows by outputting a $(4M) \times H \times W$ tensor. Here, $M$ refers to the number of flows. Therefore, every voxel flow is a 4-channel field that describes the 3-channel voxel flow $\boldsymbol{g}^i = (\boldsymbol{g}_x^i, \boldsymbol{g}_y^i, \boldsymbol{g}_z^i)$ with a corresponding weight channel $\boldsymbol{g}_w^i$. Here, $i$ $(1 \le i \le M)$ is the index of voxel flow. To synthesize the target pixels in current frame, the weights $\boldsymbol{g}_w^i$ are normalized by a softmax function across $M$ voxel

flows. We finally obtain the target pixel $\bar{\boldsymbol{x}}[x, y]$ in spatial location $[x, y]$ by calculating the weighted sum of sampling results, formulated as:

$$\bar{\boldsymbol{x}}[x, y] = \sum_{i=1}^{M} g_w^i(x, y) \boldsymbol{X}_t[x + g_x^i(x, y), y + g_y^i(x, y), g_z^i(x, y)]. \tag{1}$$

We experimentally find that compared with single voxel flow, the transmission cost of multiple voxel flows does not increase largely. The model is able to assign appropriate number of flows under the rate-distortion optimization goal. In other word, the model is optimized to avoid the transmission of unnecessary flows. Meanwhile, due to more accurate inter frame prediction, the transmission cost of residuals decreases obviously by using multiple voxel flows for weighted warping.

## 3.2 Generalized optical flow prediction

In our proposed VLVC framework, as illustrated in Fig. 2, we compress the spatial-temporal motion information via motion encoder and decoder. The concatenation of the predicted frames (*i.e.*, bilinear warping results using the predicted optical flow) and the current frame are fed into the motion encoder. The 3-D motion descriptor, *i.e.*, the voxel flows, are then decoded by the motion decoder given the quantized motion latent and the feature of predicted optical flow. In this process, the predicted optical flow reduces the spatial displacement need to be encoded, and also serves as the conditions to promote the generation of voxel flows. Thus, the optical flow prediction is clearly of great importance to reduce the redundancies of consecutive voxel flows in case of using multiple reference frames.

Specifically, there are two optical flow describe the motion between the reference frame $\hat{x}_{t_j}$ and the target frame $x_t$: $f_{t_j \to t}$ and $f_{t \to t_j}$. The flow $f_{t \to t_j}$ describe the motion of each pixel from $x_t$, and therefore we can sample $\hat{x}_{t_j}$ for each pixel in the target frame $x_t$ via bilinear (backward) warping. However, $f_{t \to t_j}$ is unknown at decoder side because the pixels of target frame is unavailable. Fortunately, the pixels from reference frames are known at both encoder and decoder. We can first estimate the optical flow of pixels from a reference frame $bol\hat{x}_{t_j}$ to other reference frames, and then predict the flow $f_{t_j \to t}$. While we obtain $f_{t_j \to t}$, it cannot be directly used for motion compensation with bilinear warping.

Recently work [31] for video interpolation proposed a forward warping method to interpolate the target frame $x_t$ by directly using the flow $f_{t_j \to t}$. For video compression, we aim to predict a approximation of the flow $f_{t \to t_j}$ to reduce the redundancies of the proposed voxel flows for better rate-distortion performance. We therefore employ the forward warping method [31] (named softmax splatting) to project the flow $f_{t_j \to t}$ to $f_{t \to t_j}$, which is a kind of flow reversal methods similar to [32]. In the following part, we will describe a novel polynomial motion modeling method to predict $f_{t_j \to t}$ given arbitrary reference frames and any target time stamp $t$. And a flow reversal layer based on softmax splatting is introduced for the final flow prediction.

**Polynomial motion modeling**   For each pixel at $t_j$, we model the motion $f_{t_j \to t}$ by the $k$-order $(k < n)$ polynomial functions:

$$f_{t_j \to t} = a_1 \times (t - t_j) + a_2 \times (t - t_j)^2 + ... + a_k \times (t - t_j)^k, \tag{2}$$

where $a_0, a_1, ..., a_k$ are the polynomial coefficients. To solve the coefficients, we set $t$ equals to the top-$k$ nearest time stamp $\{t_{j_i}\}_{i=1}^{k}$ around $t_j$ within the set of reference time stamp. Then we can obtain the following equation:

$$\begin{pmatrix} a_1 \\ a_2 \\ ... \\ a_k \end{pmatrix} = \begin{pmatrix} (t_{j_1} - t_j) & (t_{j_1} - t_j)^2 & ... & (t_{j_1} - t_j)^k \\ (t_{j_2} - t_j) & (t_{j_2} - t_j)^2 & ... & (t_{j_2} - t_j)^k \\ ... & ... & ... \\ (t_{j_k} - t_j) & (t_{j_k} - t_j)^2 & ... & (t_{j_k} - t_j)^k \end{pmatrix}^{-1} \begin{pmatrix} f_{t_j \to t_{j_1}} \\ f_{t_j \to t_{j_2}} \\ ... \\ f_{t_j \to t_{j_k}} \end{pmatrix} \tag{3}$$

where $f_{t_j \to t_{j_1}}, f_{t_j \to t_{j_2}}, ..., f_{t_j \to t_{j_k}}$ can be obtained using off-the-shelf flow estimation network. Then we can derive the polynomial coefficients and apply them to Eq. (3) predict the forward flow from $t_j$ to any time stamp $t$.

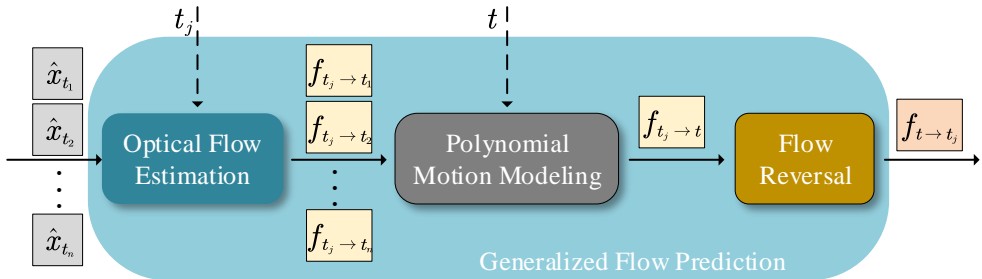

Figure 3: Generalized flow prediction module.

**Flow reversal via softmax splatting** While the forward flow $\boldsymbol{f}_{t_j \to t}$ is predicted by the polynomial functions, it cannot be directly used for motion compensation. Therefore, we introduce a flow reversal layer to forward warping $-\boldsymbol{f}_{t_j \to t}$ by softmax splatting [31]:

$$\boldsymbol{f}_{t \to t_j} = \frac{\overrightarrow{\sum}\left(\exp(\boldsymbol{Z}) \cdot (-\boldsymbol{f}_{t_j \to t}), \boldsymbol{f}_{t_j \to t}\right)}{\overrightarrow{\sum}\left(\exp(\boldsymbol{Z}), \boldsymbol{f}_{t_j \to t}\right)}, \tag{4}$$

where $\overrightarrow{\sum}$ is the summation splatting defined in [31], and $\boldsymbol{Z}$ is an importance mask generated from a small network $q$ as:

$$\boldsymbol{Z} = q(\hat{\boldsymbol{x}}_{t_j}, -\frac{1}{k}\sum_{i=1}^{k}\|\hat{\boldsymbol{x}}_{t_j} - \overleftarrow{w}(\hat{\boldsymbol{x}}_{t_i}, \boldsymbol{f}_{t_j \to t_i})\|_1), \tag{5}$$

where $\overleftarrow{w}$ is the bilinear backward warping operator.

### 3.3 Loss function

In previous works, the reference frames are determined according to pre-defined prediction modes. For example, the work of [17] applies four unidirectional reference frames, where the reference set is $\{\hat{\boldsymbol{x}}_{t-4}, \hat{\boldsymbol{x}}_{t-3}, \hat{\boldsymbol{x}}_{t-2}, \hat{\boldsymbol{x}}_{t-1}\}$. The work of [12] applies $\{(\hat{\boldsymbol{x}}_{t-1}, \hat{\boldsymbol{x}}_{t+1}), (\hat{\boldsymbol{x}}_{t-2}, \hat{\boldsymbol{x}}_{t+2}), (\hat{\boldsymbol{x}}_{t-3}, \hat{\boldsymbol{x}}_{t+3})\}$ as the reference set for bilinear prediction. In this paper, to optimize a versatile video compression model, the model will have access to various reference structures during training to adapt to different prediction modes. Therefore, we apply the loss function to cover all the frames in the entire gop as:

$$\mathcal{L} = \frac{1}{T}\sum_{t=1}^{T}[R_t(\hat{\boldsymbol{m}}_t, \hat{\boldsymbol{r}}_t | \hat{\boldsymbol{x}}_{t_i}, ..., \hat{\boldsymbol{x}}_{t_j}) + \lambda \cdot \mathcal{D}(\boldsymbol{x}_t, \hat{\boldsymbol{x}}_t | \hat{\boldsymbol{x}}_{t_i}, ..., \hat{\boldsymbol{x}}_{t_j})]. \tag{6}$$

Here, $T$ is the gop size during training. The maximum value of $T$ is seven in our experiments since a 7-frame gop can cover most prediction modes. $\{\hat{\boldsymbol{x}}_{t_i}, ..., \hat{\boldsymbol{x}}_{t_j}\}$ represents different reference set that may vary in different mini-batches. $R_t(\hat{\boldsymbol{m}}_t, \hat{\boldsymbol{r}}_t)$ is the rate of motion and residual. For simplicity, we omit the process of intra frame compression (at $t = 1$) in this loss function.

## 4 Experiments

### 4.1 Experimental setup

**Model details** The motion/residual compression modules are two auto-encoder style networks, where the bit-rate are estimated by the factorized and hyperprior entropy model [6, 7], respectively. We employ the off-the-shelf PWC-net [33] as the optical flow estimation network only in our generalized flow prediction module. We employ feature residual coding [34] instead of pixel residual coding for better performance. Detailed architecture can be found in supplementary.

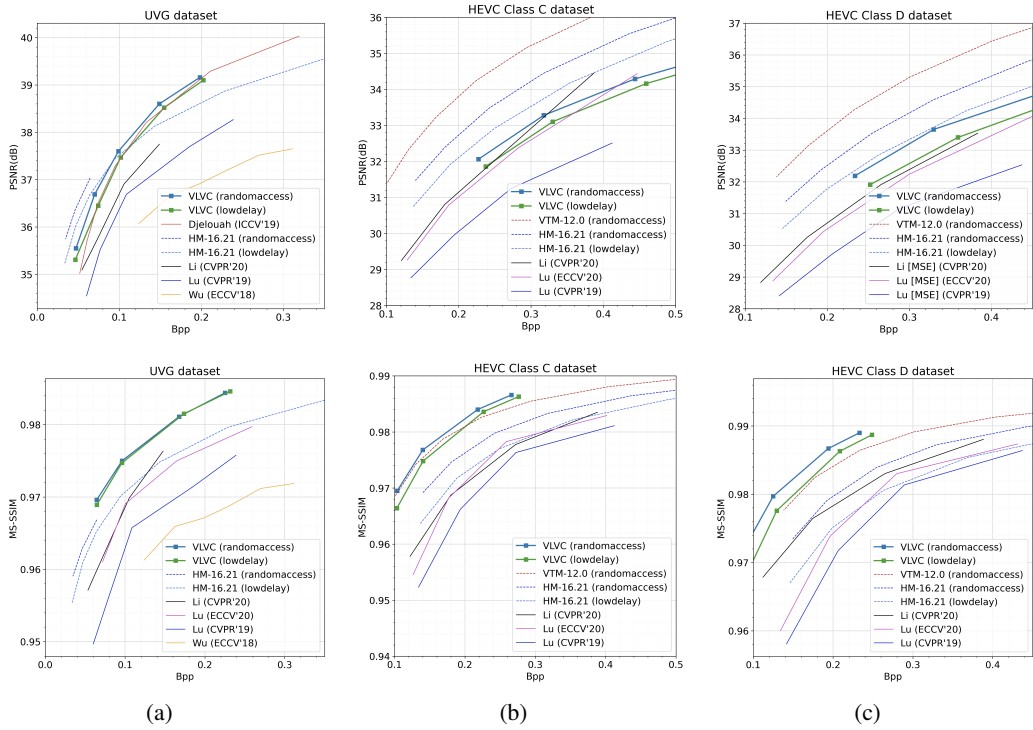

Figure 4: Rate-distortion Performance.

**Training sets**  The models were trained on the Vimeo-90k septuplets dataset [35] which consists of 89800 video clips with diverse content. The video clips are randomly cropped to $128 \times 128$ or $256 \times 256$ pixel for training.

**Testing sets**  The HEVC common test sequences [3] and the UVG dataset [36] are used for evaluation. The HEVC Classes B,C,D and E contain 16 videos with different resolution and content. The UVG dataset contains seven 1080p HD video sequences with 3900 frames in total.

**Implementation details**  We optimize four models for MSE and four models for MS-SSIM [37]. The video clip length $T$ is set to 7 for training. We use the Adam optimizer [38] with batch size of 8 and a initial learning rate of $5 \times 10^{-5}$. It is difficult to stably train the whole models from scratch. We first separately pre-train the intra-frame coding models and inter-frame coding models for MSE, with $128 \times 128$ video crops and 1,200,000 training steps. Then we jointly optimize both the models with the gop loss Eq. (6) for 100,000 steps using different metrics and $\lambda$ values. Finally, we fine-tuning all the models for $20,000$ with a crop size of $256 \times 256$ and a reduced learning rate of $1 \times 10^{-5}$

**Evaluation Setting**  We measure the quality of reconstructed frames using PSNR and MS-SSIM [37] in the RGB colorspace. The bits per pixel (bpp) is used to measure the average number of bits. We compare our method with the traditional video coding standards H.265/HEVC and H.266/VVC, as well as the state-of-art learning based methods including [15, 22, 11, 12, 17].

Recent works for learned video compression usually evaluate H.265 by using FFmpeg, with performance is much lower than official implementation. In this paper, we evaluate H.265 and H.266 by using the implementation of the standard reference software HM 16.21[39] and VTM 12.0[40], respectively. We use the default low delay and random access configuration, and modify the gop structure and key frame interval for fair comparision. Detailed configuration can be found in supplementary.

### 4.2  Performance

We evaluate our model with the state-of-the-art learned video compression approaches, including the P-frame based methods of [15, 22, 23, 17], the interpolation based methods of [11, 12]. As shown

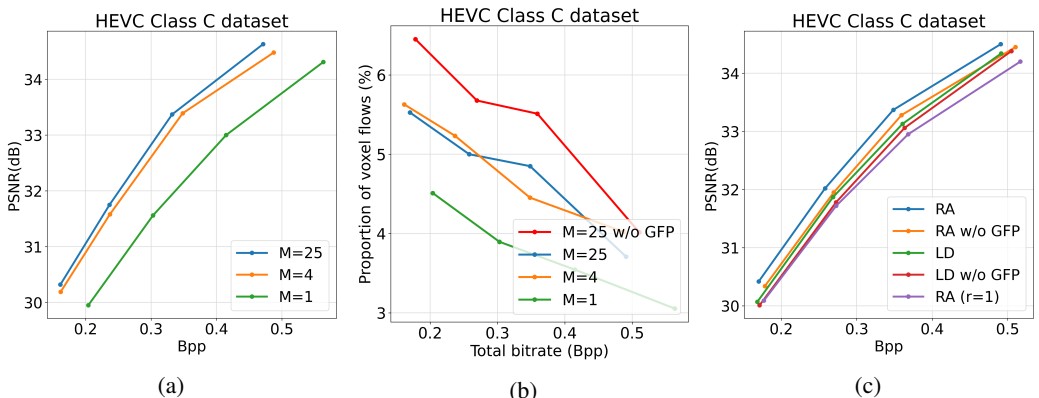

Figure 5: (a) Ablation on the number of voxel flows. (b) The Proportion of voxel flows in total bitrate. (c) Ablation on different coding configurations.

in Fig. 4, it can be observed that our proposed method significantly outperforms exiting learned video compression methods in both PSNR and MS-SSIM. Note that the *VLVC (randomaccess)* and *VLVC (lowdelay)* are two different configurations from the same models. Besides, our model is the first end-to-end learned video compression method that achieves comparable R-D performance with H.266 in terms of MS-SSIM.

## 4.3 Ablation Study and Analysis

All the models reported in ablation studies are trained for MSE using $128 \times 128$ video clips. More ablation study results and visual results can be found in the supplementary.

**The effect of the voxel flow number** As shown in Fig. 5a, the number $M$ of voxel flows significantly influence the overall rate-distortion performance. More voxel flows provide more possible sampling location for accurate motion compensation. Our proposed weighted trilinear warping with multiple voxel flows achieves about 1dB gain compared with the conventional trilinear warping with single voxel flow. Note that the performance gain is nearly saturated for $M = 25$, which is used as the default value in our models.

We also investigate the additional bitrate cost of multiple voxel flows. As shown in Fig. 5b, the proportion of multiple voxel flows in the total bitrate of video coding increases about $\frac{1}{3}$ at the same bitrate. In other words, our model can learn to improve the overall compression performance by transmitting a proper amount of additional motion information, which is represented as voxel flows.

**Versatile coding configurations** The proposed methods can deal with a various set of prediction modes. To evaluate the effectiveness of coding flexibility as well as the effectiveness of the proposed generalized flow prediction module, we simply change the input coding configurations of the same trained model at different bitrate points. Random access and low delay coding settings are denoted as *RA* and *LD*, respectively. As shown in Fig. 5c, the coding mode *RA* with bidirectional reference frames achieves a compression gain of about 0.4dB, compared with the unidirectional coding modes *LD*. Furthermore, the performance dropped about 0.1dB~0.3dB when we turn off the generalized flow prediction module for different coding settings, noted as *w/o GFP*. We also illustrate the bitrate reduction of the voxel flows shown in Fig. 5b, where *M=25* reduce the bitrate of voxel flows about $\frac{1}{6}$ compared to *M=25 w/o GFP*. Finally, we change the number of the reference frames for warping, which is set to 2 as default. We reduce the number to 1 in random access mode, noted as *RA (r=1)*, which performance is even worse than low delay setting.

**Visualization of voxel flows** The proposed voxel flows contain multiple 3-channel voxel flows $\{(\boldsymbol{g}_x^i, \boldsymbol{g}_y^i, \boldsymbol{g}_z^i)\}_{i=1}^M$ and their weights $\{\boldsymbol{g}_w^i\}_{i=1}^M$. We separately visualize the weighted temporal and spatial flow maps. The mean temporal flow map $\bar{\boldsymbol{g}}_z = \sum_i \boldsymbol{g}_w^i \cdot \boldsymbol{g}_z^i$ describes the weighted centroid of voxel flows along the time axis. As shown in the fourth column of Fig. 6a, the $\bar{\boldsymbol{g}}_z$ performs like a occlusion map for bidirectional frame prediction. The pixels in black area cannot be found in the

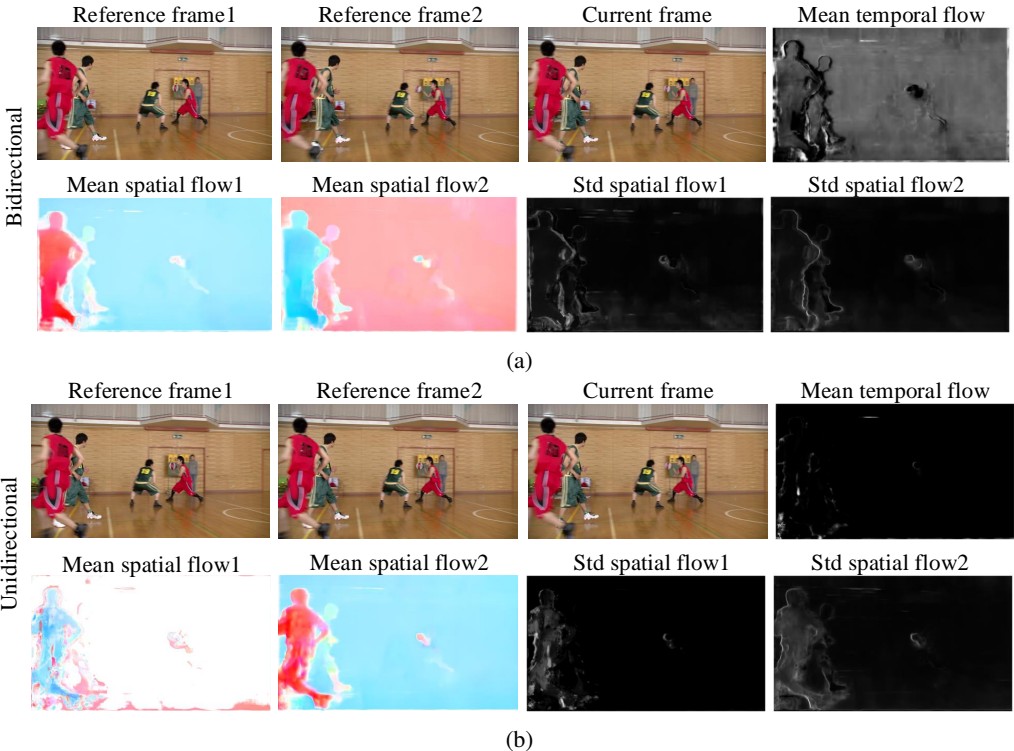

Figure 6: Visualization of the voxel flows for the same target frame with different reference frames, generated by the same model. (a) Bidirectional reference frames. (b) Unidirectional reference frames

first reference frame because the basketball player in red covers the background. Hence the voxel flows pay more attention on the second frame, resulting in large weights. The white area can be explained in the similar way for the first frame, and the gray area means that the voxel flows pay equally attention for both frames. For unidirectional frame prediction, the $\bar{g}_z$ generated by the same model are almost black everywhere, demonstrating the flexibility of trilinear warping for different prediction mode.

We also visualize the weighted mean and weighted standard deviation of spatial flow maps (noted as mean spatial flow and std spatial flow) to investigate the spatial distribution of voxel flows. We first round and group the $g_z^i$ to the nearst integer location of reference frames (e.g. 0 or 1), then separately calculate the mean flow map and std flow map for each group of voxel flows. As shown in the second and fourth raws of Fig. 6, the grouped spatial mean of voxel flows has similar distribution with optical flow. Different from optical flow, the voxel flows have large variance in the area of motion, occlusion and blur, shown in the std spatial flow map. Single optical flow is not able to find a accurate reference pixel and results in inefficient motion compensation. Multiple flow warping with weighted coefficients provide a choice to perform motion compensation using multiple reference pixels with better rate-distortion performance.

## 5 Conclusion

In this paper, we propose a versatile learned video coding (VLVC) framework that allows us to train one model to support various inter prediction modes. To this end, we apply voxel flows as a motion information descriptor along both spatial and temporal dimensions, and we perform reconstruction via proposed weighted trilinear warping using voxel flows for more effective motion compensation. Through formulating different inter prediction modes by a unified polynomial function, we design a novel flow prediction module to predict accurate motion trajectories. In this way, we significantly reduce the bits cost of encoding motion information. Thanks to above novel motion compensation and flow prediction, VLVC not only achieve the support of different inter prediction modes but also

yield competitive R-D performance compared to conventional VVC standard, which fosters practical
applications of learned video compression technologies.

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
