# Supplementary materials: Versatile Learned Video Compression

## 1 Performance

### 1.1 Configurations of the HEVC/VVC reference software

We first convert the source video frames from YUV420 to RGB by using the command:

$$\text{ffmpeg} \, \text{--r} \, [FPS] \, \text{--s} \, [W] * [H] \, \text{--pix\_fmt yuv420p --i} \, [IN].\text{yuv} \, [OUT].\text{png}$$

where *FPS* is the frame rate, *W* is width, *H* is height, *IN* is the name of input file and *OUT* is the name of output file. As mentioned in [1], it is not ideal to evaluate the standard codecs in RGB color space because the native format of test sets are YUV420. To reduce this effect, we treat the RGB video frames as the source data and convert them into YUV444 as the input of the standard codecs. The reconstructed videos are converted back into RGB for evaluation. This kind of operation is commonly used in recent works of learned image compression [2, 3].

**HEVC reference software (HM)** For lowdelay setting, we simply use the default *encoder_lowdelay_P_main.cfg* configuration file of HM 16.21 [4]. For randomaccess setting, we change the gop structure of the default *encoder_randomaccess_main.cfg* configuration file as following:

```
#       Type POC QPoffset QPOffsetModelOff QPOffsetModelScale CbQPoffset CrQPoffset QPfactor tcOffsetDiv2 betaOffsetDiv2
Frame1:  P   6   1     0.0              0.0                  0          0          1.0       0            0
Frame2:  B   3   4    -5.0              0.2                  0          0          1.0       0            0
Frame3:  B   2   5    -6.0              0.25                 0          0          1.0       0            0
Frame4:  B   1   6    -7.0              0.3                  0          0          1.0       0            0
Frame5:  B   4   5    -6.0              0.25                 0          0          1.0       0            0
Frame6:  B   5   6    -7.0              0.3                  0          0          1.0       0            0

         temporal_id #ref_pics_active #ref_pics reference pictures    predict   deltaRIdx-1
             0             1               1        -6                    0          0
             1             2               2        -3  3                 2          0
             2             2               3        -2  1  4              2          0
             3             2               4        -1  1  2  5           2          0
             2             2               3        -4 -1  2              2          0
             3             2               4        -5 -2 -1  1           2          0
```

Submitted to 35th Conference on Neural Information Processing Systems (NeurIPS 2021). Do not distribute.

14  The following command is used to encode all HM videos:

TAppEncoderStatic –c [CFG] –i [IN].yuv –b [OUT].bin –o [OUT].yuv –wdt [W] –hgt [H]
–fr [FPS] –f [N] –q [QP] --IntraPeriod=12 --Profile=main_444
--InputChromaFormat=444 --Level=6.1
--ConformanceWindowMode=1

15  where $N$ is the number of frames to be encoded for each sequence, which is set as 100 for the HEVC
16  dataset and 600 for the UVG dataset.

17  **VVC reference software (VTM)**  For randomaccess setting, we change the gop structure of the
18  default *encoder_randomaccess_main.cfg* configuration file of VTM 12.0 [5] as following:

| # | Type | POC | QPoffset | QPOffsetModelOff | QPOffsetModelScale | CbQPoffset | CrQPoffset | QPfactor | tcOffsetDiv2 | betaOffsetDiv2 | CbTcOffsetDiv2 | CbBetaOffsetDiv2 | CrTcOffsetDiv2 |
|---|---|---|---|---|---|---|---|---|---|---|---|---|---|
| Frame1: | P | 6 | 1 | 0.0 | 0.0 | 0 | 0 | 1.0 | 0 | 0 | 0 | 0 | 0 |
| Frame2: | B | 3 | 4 | -5.0 | 0.2 | 0 | 0 | 1.0 | 0 | 0 | 0 | 0 | 0 |
| Frame3: | B | 2 | 5 | -6.0 | 0.25 | 0 | 0 | 1.0 | 0 | 0 | 0 | 0 | 0 |
| Frame4: | B | 1 | 6 | -7.0 | 0.3 | 0 | 0 | 1.0 | 0 | 0 | 0 | 0 | 0 |
| Frame5: | B | 4 | 5 | -6.0 | 0.25 | 0 | 0 | 1.0 | 0 | 0 | 0 | 0 | 0 |
| Frame6: | B | 5 | 6 | -7.0 | 0.3 | 0 | 0 | 1.0 | 0 | 0 | 0 | 0 | 0 |

| CrBetaOffsetDiv2 | temporal_id | #ref_pics_active_L0 | #ref_pics_L0 | reference_pictures_L0 | #ref_pics_active_L1 | #ref_pics_L1 | reference_pictures_L1 |
|---|---|---|---|---|---|---|---|
| 0 | 0 | 1 | 1 | 6 | 0 | 0 | 0 |
| 0 | 1 | 1 | 1 | 3 | 1 | 1 | -3 |
| 0 | 2 | 2 | 2 | 2 -1 | 2 | 2 | -1 -4 |
| 0 | 3 | 2 | 2 | 1 -1 | 2 | 3 | -1 -2 -5 |
| 0 | 2 | 2 | 3 | 4 2 1 | 2 | 2 | 1 -2 |
| 0 | 3 | 2 | 3 | 5 2 1 | 2 | 2 | 1 -1 |

19  The following command is used to encode all VTM videos:

EncoderAppStatic –c [CFG] –i [IN].yuv –b [OUT].bin –o [OUT].yuv –wdt [W] –hgt [H]
–fr [FPS] –f [N] –q [QP] --IntraPeriod=12 –c yuv444.cfg
--InputBitDepth=8 --OutputBitDepth=8
--InputChromaFormat=444 --Level=6.1
--ConformanceWindowMode=1

20  where $N$ is the number of frames to be encoded for each sequence, which is set as 100 for the HEVC
21  dataset and 600 for the UVG dataset.

22  **1.2  Model Complexity.**

23  The total size of our inter-frame compression model is about 103MB, where the off-the-shelf optical
24  flow estimation network (PWC-net [6]) takes about 36MB. We use the 1080p videos to evaluate the
25  encoding/decoding time with one 2080TI GPU (11GB memory) and one Intel(R) Xeon(R) Gold 5118
26  CPU @ 2.30GHz. VLVC runs at 1587ms/frame for encoding and 1471ms/frame for decoding. The
27  portion of arithmetic entropy coding (on CPU) takes about 70% of the total runtime.

28  **1.3  R-D curves on the HEVC Class B and Class E datasets.**

29  We also compare against VVC on the datasets of UVG, HEVC ClassB and HEVC ClassE, as shown
30  in Fig. 1. Compared with VVC, our method performs worser in low bit-rate but better in high bit-rate
31  when evaluated by MS-SSIM. The learning-based codecs Li(CVPR'20) [7], Lu(ECCV'20) [8],
32  Lu(CVPR'19) [9] and Wu(ECCV'18) [10] are also included for comparision.

33  **1.4  BD-rate**

34  In table 1, we provide the BD-rate [11] savings of VLVC (randomaccess) relative to VVC (rando-
35  maccess) in terms of MS-SSIM. Our proposed VLVC saves more bit-rate than VVC on various
36  benchmark datasets.

| | UVG | Class B | Class C | Class D | Class E |
|---|---|---|---|---|---|
| VLVC | -0.97% | -4.71% | -7.37% | -18.25% | -6.31% |

Table 1: BD-rate savings of VLVC relative to VVC in terms of MS-SSIM on different datasets. Negative values indicate BD-rate savings.

## 2 Architecture Details

In Fig. 2 and Fig. 3 we show the detailed architecture of our models. For motion compression, we employ the factorized density model [2] to estimate the entropy of quantized motion latents. For residual compression, following the work of [12], we build a network of feature residual coding with a modified version of the hyperprior model [2, 3]. The detailed structure of the deployed hyperprior model can be found in [12].

## 3 Subjective Comparison

To verify if high MS-SSIM scores lead to high subjective quality in our models, we visualize the reconstruction of VLVC and VVC with similar average bitrate on the HEVC ClassB dataset (0.1945 bpp and 0.2238 bpp, respectively). As shown in Fig. 4 and Fig. 5, our reconstruction has better subjective quality than VVC.

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

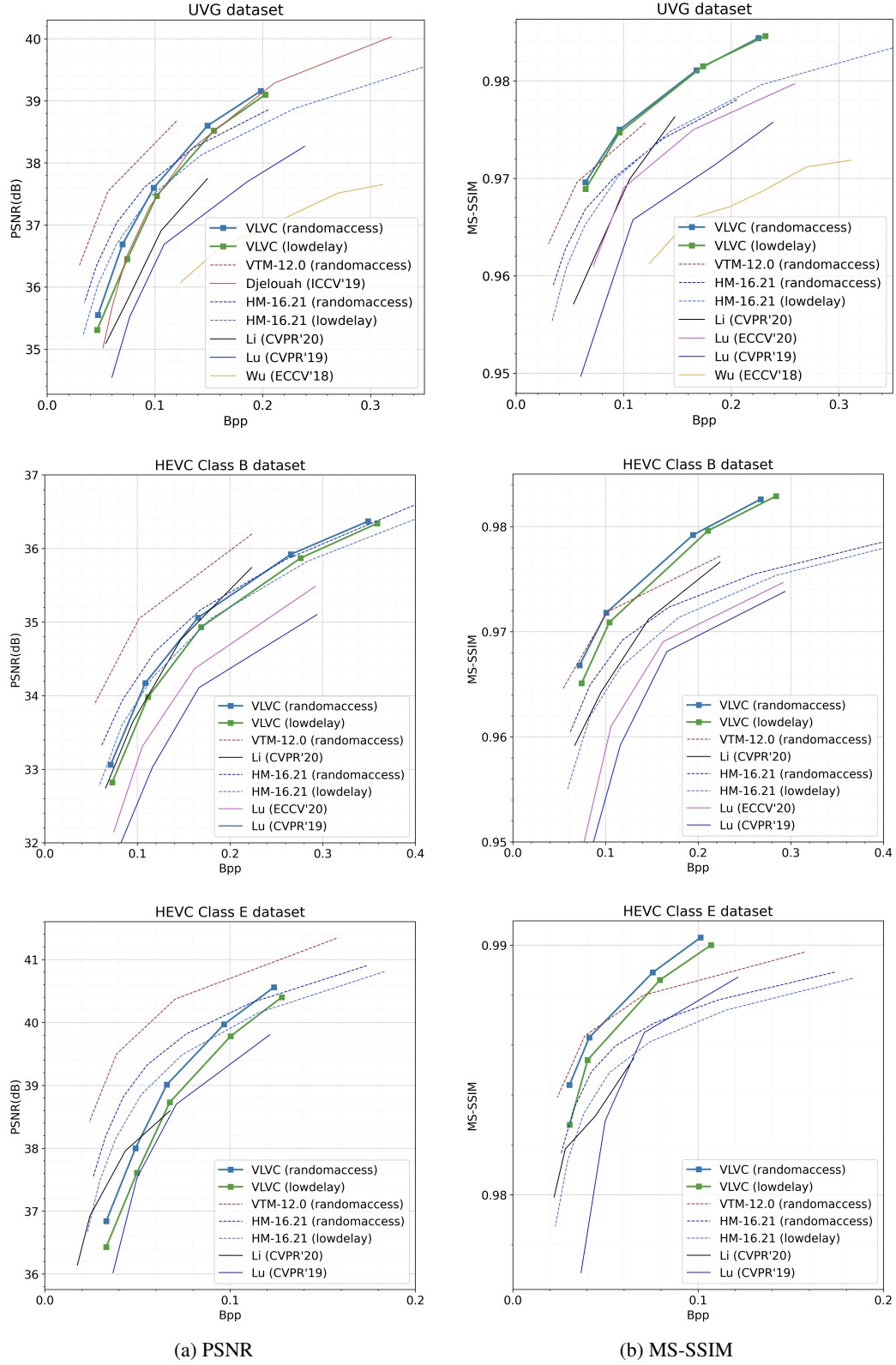

(a) PSNR

(b) MS-SSIM

Figure 1: Rate-distortion curves on the UVG, HEVC Class B and Class E datasets.

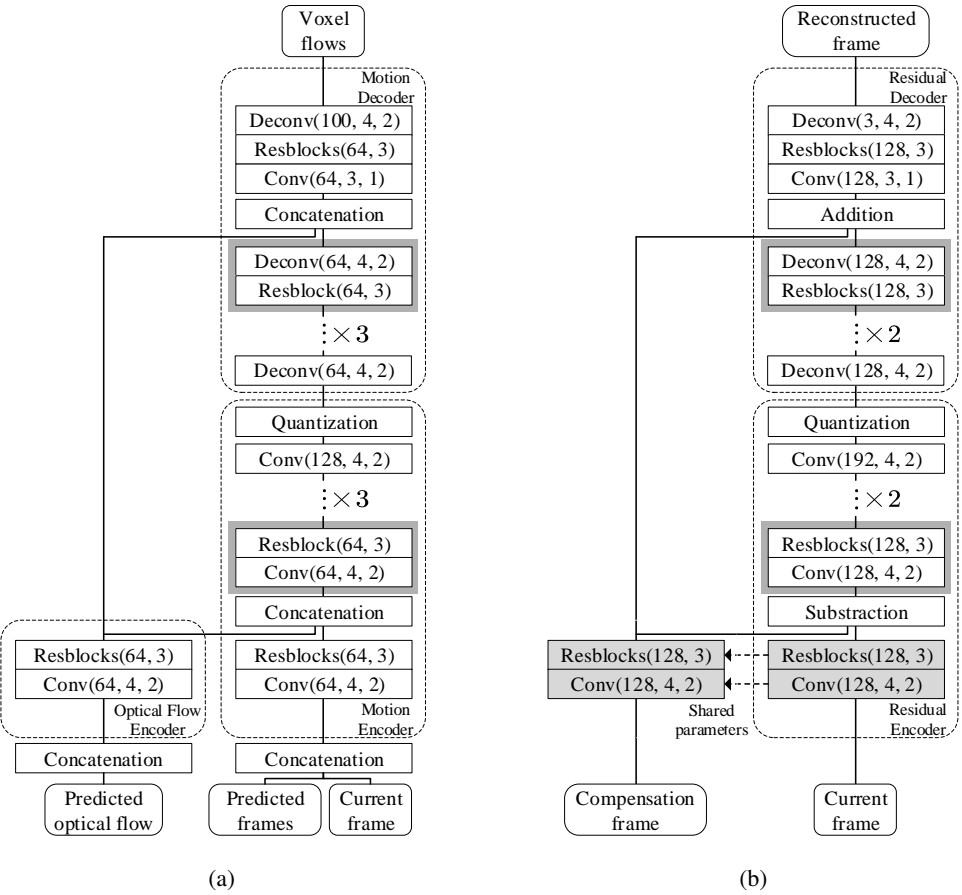

Figure 2: Detailed structure of (a) the Motion Encoder/Decoder and the Optical Flow Encoder (b) the Residual Encoder/Decoder. *Conv(C, K, S)* and *Deconv(C, K, S)* represent the convolution and deconvolution layers with C output channels and a kernel size of K and a stride of S. The details of *Resblock* and *Resblocks* are shown in Fig. 3.

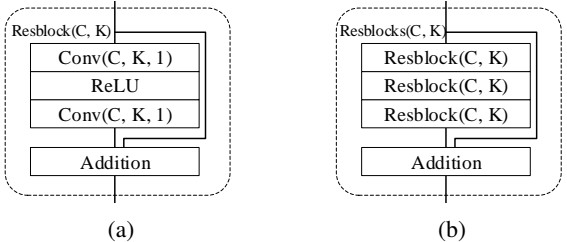

Figure 3: Detailed structure of the *Resblock* and *Resblocks*. *Conv(C, K, S)* represents the convolution layer with C output channels and a kernel size of K and a stride of S.

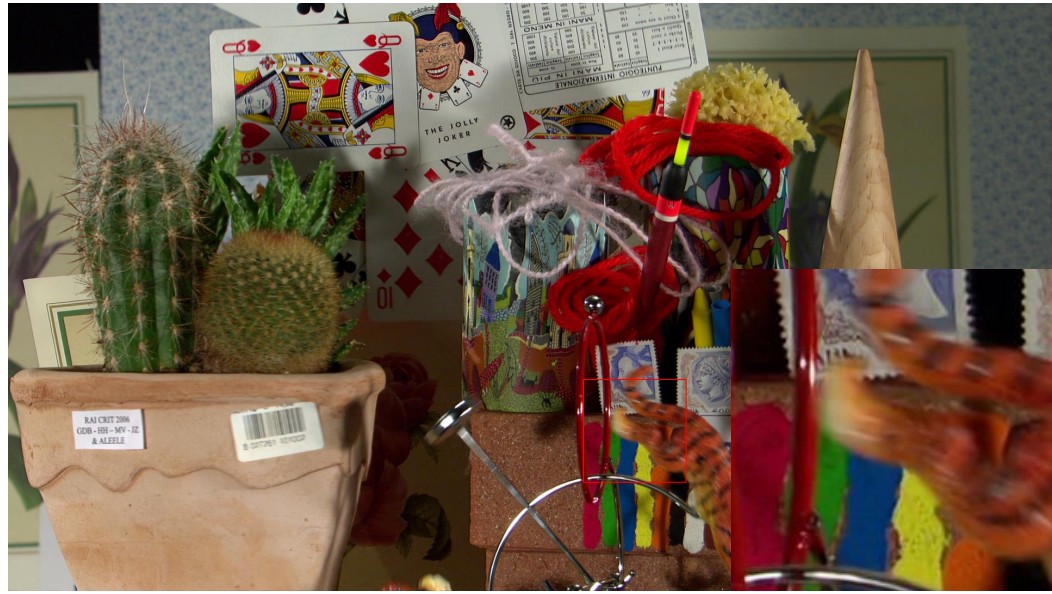

VLVC, PSNR(dB)/MS-SSIM: 33.21/0.9734

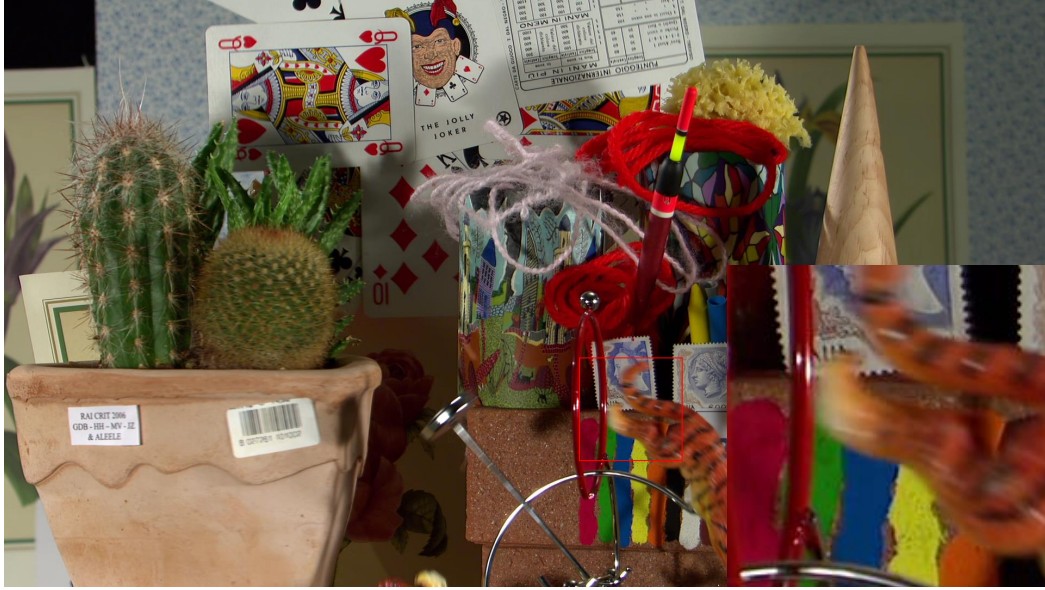

VVC, PSNR(dB)/MS-SSIM: 34.64/0.9692

Figure 4: Subjective comparison between our proposed VLVC and VVC on a reconstructed frame of the video 'Cactus' in HEVC ClassB. The reconstructed frame of VLVC is sharper and richer in texture while the average bpp is smaller.

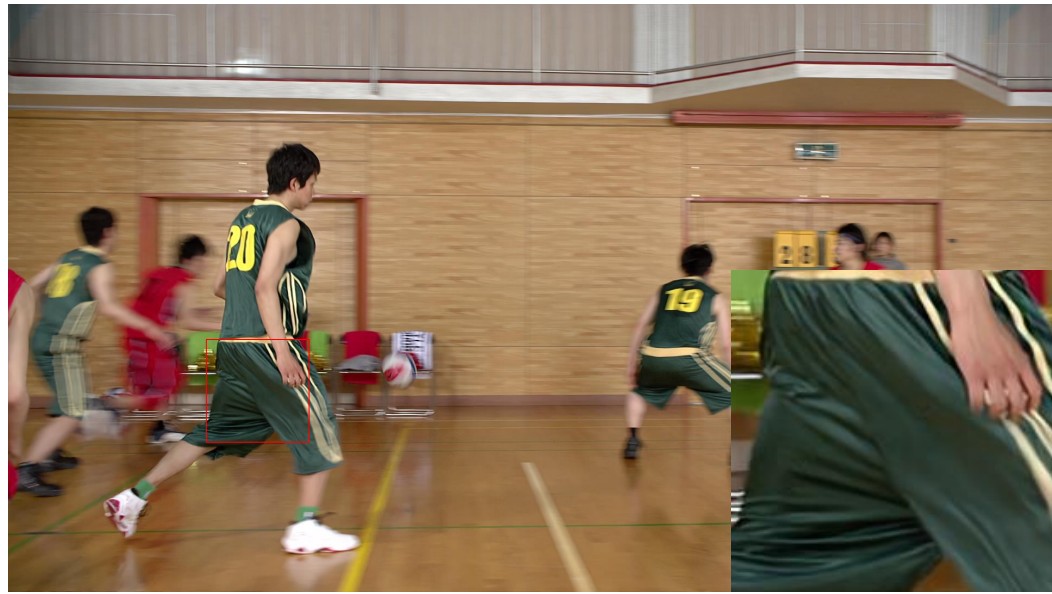

VLVC, PSNR(dB)/MS-SSIM: 35.27/0.9796

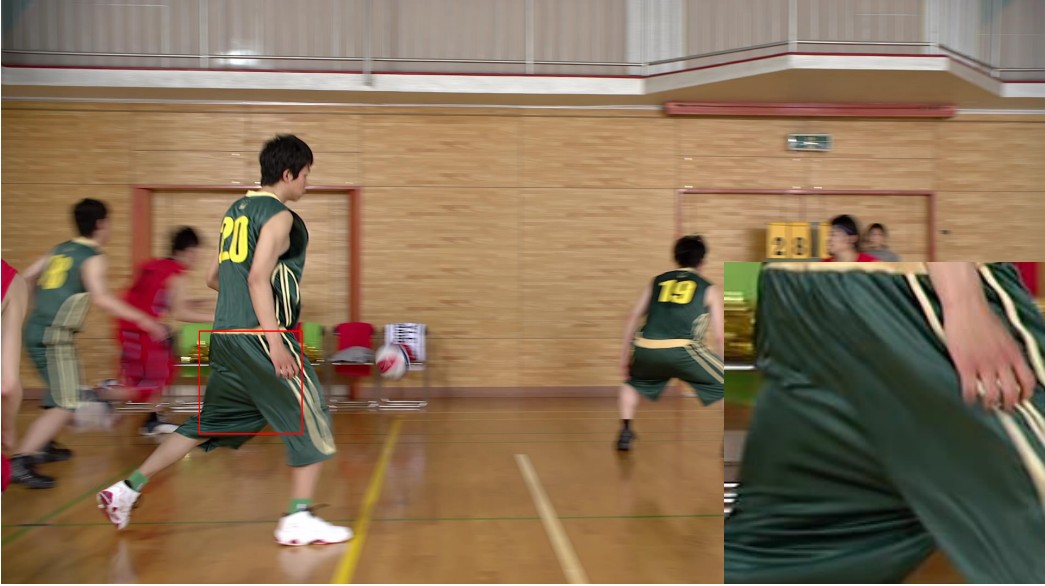

VVC, PSNR(dB)/MS-SSIM: 36.81/0.9778

Figure 5: Subjective comparison between our proposed VLVC and VVC on a reconstructed frame of the video 'BasketballDrive' in HEVC ClassB. The reconstructed frame of VLVC is sharper and richer in texture and while the average bpp is smaller.