# OpenReview forum: "Versatile Learned Video Compression"
_NeurIPS.cc/2021/Conference — NeurIPS 2021 Submitted_

### Official Review · Reviewer_Eb8Q · 2021-07-03

**Rating:** 6
**Confidence:** 4

**Summary:**

First, the paper introduces the idea of using multiple predictions (e.g. M=25) for learning-based video coding. Second, to reduce the required motion bits, a flow prediction method using polynomial approximation is developed. The idea is reasonable, and the reported performance seems promising. However, the paper is written rather poorly, and how the algorithm is actually designed is not clear. Thus, the reported performance is not convincing either.

**Main Review:**

***
This paper is slightly longer than the page limit (9 pages). Anyway, it violates the rule and can be rejected. I leave the decision to AC.

***

Except for the survey of previous work, the writing and description are poor. There are many typos and grammatical errors. For example,
1. gop -> GOP
2. More attention should be paid to articles (a, the) and tenses.
3. "Considering a practical scene where multiple objects of the same types appear at the same time, more than one areas could
 referred for reconstructing the local region." What does "the local region" specify?
4. More possible sampling location
5. "$D$ is the number of reference frames used for motion compensation." Motion estimation? You use only 2 frames for motion compensation. Is that right?
6. In this process, the predicted optical flow reduces the spatial displacement need to be encoded
7. there are two optical flow describe the motion between the reference frame and  the target frame
8. the flow describe
9. pixels is unavailable.

In fact, there are too many errors to list them all, especially near the end of the paper.





There are many technical details, which cannot be clearly understood even after reading the supplemental document.
1. Please describe the prediction structure for random access at $r=2$ when the GOP size is 7.
2. When there are only two reference frames ($r=2$), what is the role of $g_z$ ? Why do you need it anyway? You are using multiple voxel flows. So half of them can be constrained from reference 1, and the other from reference 2.
3. Figure 6 is hard to understand. Please provide the color bars.
4. How do you encode intra frames? Using the same residual encoder? Are they considered in the rate-distortion curves?
5. How do you set $k$ in Eq. (2) and (3)? Is a higher-order polynomial (beyond quadratic) really necessary? Also, here, the variable is $\Delta t = t_j - t_i$. Is it ok to ignore the distance from the current frame $t$?







**Time Spent Reviewing:**

3

---

> ### Author Response · Authors · 2021-08-10
> **Authors' response**
>
> Thanks for your comments. We will carefully refine our paper to avoid typos in our revised version. However, it seems that you do not fully understand our work. Hope for more discussions with you to address your concerns and doubts. Regarding your questions, our answers are as below:
>
> >**Q1**: Please describe the prediction structure for random access at $r=2$ when the GOP size is 7.
>
> **A1**: The prediction structure is almost the same as the GOP structure of VTM described in the supplementary material. We choose the nearest 2 reference frames for the voxel flow based motion compensation, and the number of reference frames for flow prediction is set as 3.
>
>
> >**Q2**: When there are only two reference frames ($r=2$), what is the role of $g_z$ ? Why do you need it anyway? You are using multiple voxel flows. So half of them can be constrained from reference 1, and the other from reference 2.
>
> **A2**: The axis $g_z$ indicates the temporal sampling position for voxel flow based warping. Such design enables our model to adaptively choose appropriate reference frames according to the video content. Thus, more flows will be assigned to the reference frame that is more relative to the current frame, which is demonstrated by our experiments. For the bidirectional case in Figure 6a, around half of the voxel flows are directing to the previous reference frame, while the other half are directing to the following reference frame. For the unidirectional case, if two previous reference frames are available, usually our model will assign more flows to the closer reference frame. Specifically, for the example in Figure 6b, among all the 25 flows, about 20 flows are directing to the closer frame (noted as reference frame 2 in this figure). This is why our proposed weighted voxel flows are flexible to achieve better prediction results.
>
> >**Q3**: Figure 6 is hard to understand. Please provide the color bars.
>
> **A3**:  Thanks for your suggestions. We will add the description of the color bars in the revision. For the "Mean spatial flow", like optical flow, different color represents different motion direction. For “Std spatial flow”, different colors represent the difference among multiple flows, whereas the white color represents high std value. For "Mean temporal flow", different color means different temporal reference position, where the white color means $g_z$  is close to reference 1 and the black color means $g_z$ is close to reference 2.
>
>
> >**Q4**: How do you encode intra frames? Using the same residual encoder? Are they considered in the rate-distortion curves?
>
> **A4**: The I-frames codec is also a learning-based model with hyperprior-based coding structure [1]. We replace the GDN with residual blocks. The bitrate of intra-frame is of course considered in the rate-distortion curves.
>
> >**Q5**: How do you set $k$ in Eq. (2) and (3)? Is a higher-order polynomial (beyond quadratic) really necessary? Also, here, the variable is $\Delta t = t_i - t_j$ . Is it ok to ignore the distance from the current frame?
>
> **A5**: $k$ is usually set to 2 for computation efficiency. A high-order polynomial model (beyond quadratic) is not necessary for the most use case. Please note that the polynomial motion model is mainly designed for dealing with various reference configurations for versatile video compression, which is under-explored in previous works. The variable $ \Delta t = t_i - t_j $ appears in Eq.(3), used for solving the polynomial coefficients in Eq.(2). And the distance $ t - t_j $ is considered in Eq.(2) for predicting the forward flow based on the polynomial coefficients.
>
>
> **In addition**, we did not notice that our final modification before the submission deadline made the paper exceed the page limit, which was caused by the Latex automatic compiling. We hope it will not affect the technical decision of our paper. We appreciate your efforts and are looking forwards to your kind understanding for the format issue.
>
> Reference:
> - [1] Variational image compression with a scale hyperprior, ICLR 2018.

---

> > ### Comment · Reviewer_Eb8Q · 2021-08-28
> > **Thanks for the clarifications.**
> >
> > They are good enough, and I will upgrade my rating.

---

> > > ### Author Response · Authors · 2021-08-29
> > > **Authors' response**
> > >
> > > Thanks for your positive comments and helpful suggestions.

---

> ### Author Response · Authors · 2021-08-21
> **Authors' response**
>
> Many thanks for your efforts in reviewing. We are looking forward to know whether our responses well address your concerns. Please feel free to reach out to us if you have further suggestions.

---

### Official Review · Reviewer_hMDd · 2021-07-13

**Rating:** 6
**Confidence:** 4

**Summary:**

The authors introduce VLVC, a new neural video compression codec. Thanks to a new 3D motion compensation structure based on spatio-temporal interpolation, a single trained model can be run in arbitrary prediction modes (such as low-delay or random-access settings). In addition, each frame can be compressed with an arbitrary number of reference frames. There are a number of smaller changes to established neural video codecs. The resulting VLVC codec is demonstrated on standard datasets, showing good rate-distortion results (in particular when trained and evaluated on MS-SSIM).

**Ethical Concerns:**

I see no ethical issues with this paper.

**Limitations And Societal Impact:**

The authors do not discuss the societal impact that neural video codecs could have. The paper could be improved by discussing risks for instance from biases in the training data, but also due to algorithmic choices.

**Main Review:**

Neural video compression is without question a problem of large practical importance. It seems reasonable that a NN codec that can operate both in low-delay and random-access modes is beneficial, though I find that the authors overstate this case. Blanket statements like "existing learned video compression schemes [...] are inapplicable for various scenarios" really need to be supported by more arguments or examples.

VLVC brings novelty to the neural video compression landscape. The independence of a model from the prediction mode is made possible by a new warping algorithm based on ultiple 3D voxel flows and spatio-temporal polynomial interpolation. However, I believe this is not the first work in this direction: T. Ladune et al [1] also propose a framework in which a single model can be used to compress frames in different ways. A comparison of these methods would enhance the paper.

The proposed method is overall sound, as far as I can tell. In addition to the flexibility with respect to coding scheme / reference structure, the structure with multiple flows allows the model to deal better with occlusions and to model motion uncertainty to some extent, both of which may be relevant in practice.

Some aspects I found confusing; in particular, I am not sure if I understood the generalized optical flow prediction (section 3.2) correctly. Perhaps more importantly, especially for a paper that focuses so much on the independence between reference structures: which reference configurations are actually used? In line 216 the authors describe it as "the model will have access to various reference structures" without going into detail. The labels "randomaccess" (RA) and "lowdelay" (LD) also do not fully specify this. If there was a definition of these somewhere in the paper, I missed it and would very much appreciate a pointer. Are the reference structures described for HM and VTM in the supplementary material used for VLVC as well? How are I-frames / key frames handled in this approach?

I find that several aspects of this approach deserve more discussion. For instance:
1. When should one use which coding configuration (reference structure)? Can the model be extended to make such decisions dynamically based on video content?
2. What exactly are the benefits of having a single model that supports arbitrary coding configurations? Can the authors for instance argue or demonstrate that with the same overall computational budget, a common model achieves a better performance than separate LD and RA models?
3. How fast is the proposed video codec during training, encoding, and decoding relative to other neural video codecs? How efficient is the new 3D voxel warping operation? The supplementary material mentions encoding and decoding around 0.7 fps, can the authors further comment on how this could be improved?
4. It would be interesting to have a detailed comparison of the proposed 3D warping and the scale-space flow (SSF) warping proposed by E. Agustsson et al [2]. Both methods use 3D flows, although the third dimension in SSF refers to a blur direction rather than time. Both have the interesting feature that they improve the modelling of uncertainty in the optical flow, for instance in the presence of movement or occlusion (see Fig. 6). Can the authors comment more on the similarity and differences between these approaches?
5. How important is the off-the-shelf PWC-Net in this approach? How does the use of this pretrained optical flow estimator fit together with Sec. 3, where the authors argue that end-to-end training with implicit flow networks  performs better than pretrained optical flow networks and simplifies the network architecture?

The empirical results show a strong rate-distortion performance on standard datasets, especially in the low-bitrate region. Comparing to HM and VTM is laudable, though I do not see why the authors changed the default RA reference structure in these classical codecs. The authors also compare to several neural baselines. Some strong baselines like SSF [2], ELF-VC [3], and B-EPIC [4] are missing (except for SSF these all came out very recently though).

While the results are strong, they do not quite justify all of the claims that the authors make. VLVC does not "yield competitive R-D performance compared to [...] VVC" (last sentence of the conclusions) when evaluated on PSNR, i.e. the metric that classical codecs are largely optimized for. I also do not see that the "proposed method significantly outperforms existing learned video compression methods" when the strongest neural baselines are missing from the comparison. (I tried to compare VLVC to SSF, ELF-VC, and B-EPIC manually and it does seem to me that VLVC outperforms them slightly at low bitrates! But the authors need to show this to make SOTA claims). I wonder why the authors hide the meaningful classes B and E of the HEVC test datasets in the supplementary material, while showing results for classes C and D with their extremely small resolutions (HEVC class D has a resolution of 240x416...). Finally, it would be great to provide more BD-rate savings (for all considered datasets and also for the MSE models).

The authors present a good set of ablations and extensive supplementary materials. The main ablation I am missing is a comparison to the same method when it is trained on only one prediction mode.

The presentation of the paper could be improved. I believe the explanations could be clearer, important aspects could be highlighted better, and (as discussed above) several aspects seem to be missing. There are also several typos.

All in all, the authors present an interesting neural video compression method with several new features and a strong empirical performance. Unfortunately, this paper does not do the method justice. Both the description of the method, the discussion of its properties and use cases, and the empirical evaluation can be improved in several ways. A revised version that answers my question and addresses these points might make a good paper, but in its current form I do not think it is ready to be accepted.

References:
- [1] T. Ladune et al, Conditional coding for flexible learned video compression, ICLR 2021 workshop neural compression
- [2] E. Agustsson et al, Scale-space flow for end-to-end optimized video compression, CVPR 2020
- [3] O. Rippel et al, ELF-VC: Efficient Learned Flexible-Rate Video Coding, arXiv:2104.14335
- [4] R. Pourreza and T. Cohen, Extending Neural P-frame Codecs for B-frame Coding, arXiv:2104.00531


**Time Spent Reviewing:**

5

---

> ### Author Response · Authors · 2021-08-10
> **Authors' response**
>
> >**Q1**: About the coding configurations of VLVC.
>
> **A1**: Sorry for this confusion. We will add a more detailed introduction about our coding configurations in the revision. In training stage, for each iteration we randomly select a coding order in the set of {(0, 1, 2, 3, 4, 5, 6), (0, 6, 3, 2, 1, 4, 5), (0, 6, 3, 1, 2, 5, 4)}, where the number indicates the frame index. During training, the structure and number of reference frames are randomly chosen in each iteration. In the testing stage, the GOP structure of VLVC in random access setting is almost the same as the reference structures described for HM and VTM in the supplementary material. The default low delay setting of VLVC is similar to the low delay B setting of HM, where the flow prediction module is turned on and the number of reference frames is set as 3.
>
> >**Q2**: How are I-frames handled in this approach?
>
> **A2**: The model used for I frame coding is an autoencoder-based model with hyperprior-based coding structure [1]. We replace the GDN layer with residual blocks. We will clearly describe this detail in our revision. Thanks for your question.
>
> >**Q3**: When should one use which coding configuration? Can the model be extended to make such decisions dynamically based on video content?
>
> **A3**:  Similar to traditional video codecs, using which coding configuration depends on the demands of application scenarios. For example, the low-delay mode is suitable for the scenarios like live streaming, where an input video frame should be immediately compressed and transmitted. Random-access mode is suitable for the scenarios like playback. Usually, the coding configuration is manually determined like traditional codecs.
>
> >**Q4**: The benefits of having a single model that supports arbitrary coding configurations. Can the authors for instance argue or demonstrate that with the same overall computational budget, a common model achieves a better performance than separate LD and RA models?
>
> **A4**:  For practical applications, it’s of very high importance to make the codec support different coding configurations, and this is true for all traditional codecs in use currently. Having a single model that supports arbitrary coding configurations not only provides a unified video coding framework but also reduces the model size.  For VLVC, a unified model optimized for arbitrary coding configurations performs slightly worse than the separately optimized LD and RA models, but the model size is 2x smaller. More importantly, VLVC supports more scalable coding configurations (the number and location of reference frames, the use of flow prediction module, ...) in a single model, compared with existing learning-based codecs.
>
> >**Q5**: How fast is the proposed video codec? How efficient is the new 3D voxel warping operation? The supplementary material mentions encoding and decoding around 0.7 fps, can the authors further comment on how this could be improved?
>
> **A5**:  We will add more details about the complexities in the revision.
>
> The training of VLVC consists of three parts: I-frame codec pretraining (1 day on a 2080Ti GPU), inter-frame codec pretraining (2 days on a 2080Ti GPU) and joint training (2 days on four 2080Ti GPUs). The training time is comparable with recent works like SSF [2] (4 days on a NVidia V100 GPU), DVC_pro [3] (4 days on two GTX 1080Ti GPUs) and B-EPIC [7] (10 days on a Nvidia V100 GPU).
>
> The overall runtime of VLVC is comparable with recent learning-based codecs, such as DVC [1] (single reference frame) and MLVC [2] (multiple reference frames). For a fair comparison, we reproduce the works of DVC and MLVC using PyTorch and only compare the network inference time on 1080p videos, except for the time of arithmetic coding (on CPU). As shown in the table, the VLVC (LDP), VLVC (LDB) and VLVC (RA) are low delay P (unidirectional, single reference frame), low delay B (unidirectional, multiple reference frames) and random access (bidirectional, multiple reference frames) modes of VLVC, respectively. The runtime of our method is slightly less than DVC and MLVC under similar coding modes. The time of arithmetic coding is not included because it is a common part of any codec and is sensitive to implementation.
>
> |    | DVC | VLVC (LDP) | MLVC | VLVC (LDB) | VLVC (RA) |
> |  :---:   |  :---:  |  :---:  |  :---:  |  :---:  |  :---:  |
> | encoding time (s) | 0.59 | 0.46 | 1.23 | 0.86 | 0.86 |
> | decoding time (s) | 0.32 | 0.33 | 0.99 | 0.74 | 0.74 |
>
> The new 3D voxel warping operation runs at 0.033s per frame for HD 1080.
>
> To further improve the encoding/decoding time, one way is to construct a data buffer that stores the previously estimated optical flow or the intermediate features of the optical flow network. A decoded frame (especially I-frame) is usually served as the reference for many other frames. Thus, the data buffer can significantly improve the inference speed of the generalized flow prediction module. Another way is to optimize and accelerate the part of arithmetic entropy coding using parallel computing techniques, like ELF-VC [6].
>
> >**Q6**: The similarity and difference between our VLVC and SSF [2]?
>
> **A6** (1) About difference. In SSF in order to smooth the warping result, Gaussian smoothing is applied, where the smoothing kernel shape and kernel weights are restricted in a predefined set (square Gaussian kernels). However, in VLVC, we adopt more flexible motion description via the weighted warping of multiple flows, compared with a deterministic flow adopted in SSF. (2) About similarity. It is interesting to find that the variance of multiple flows in our method, which is visualized in Figure 6, has the similar distribution to the scale axis in SSF. This result reveals that both SSF and our proposed weighted voxel flows are advantageous for modeling motion in an uncertain manner, such as fast movement or occlusion.
>
> >**Q7**: The importance of the off-the-shelf PWC-Net in our method.
>
> **A7**: In our model, the off-the-shelf PWC-Net is only used as a part of the flow prediction module, which is turned off in the case of single reference frame. Our main voxel flow encoder and decoder are analogous to the VAE-based image compression model and do not contain a pretrained flow extractor. In short, when we predict with multiple reference frames, the PWC-Net is necessary to predict optical flow as the auxiliary condition that facilitates the generation and compression of voxel flows. But when we predict with single reference frame, we do not require the PWC-Net and turn off the entire flow prediction module.
>
> >**Q8**: Why the authors changed the default RA reference structure in these classical codecs.
>
> **A8**: The default RA GOP size in HM and VTM is 16 or 32, larger than the intra-frame interval (10 or 12) used for evaluation in recent learning-based codecs. For a fair comparison, we try to keep the intra-frame interval consistent in our model and traditional codecs and therefore change the GOP structure.
>
> >**Q9** Comparison to SSF, ELF-VC, and B-EPIC.
>
> **A9**: ELF-VC [6] and B-EPIC [7] are two concurrent works and have not been published yet. We therefore didn't include them for comparison in the main paper. Here we provide the BD-rate savings of VLVC against SSF, ELF-VC and B-EPIC in terms of MS-SSIM.
>
> |  Codec  | UVG | MCL-JCV | ClassB | ClassC | ClassD | ClassE |
> |  :---   |  ---:  |  ---:  |  ---:  |  ---:  |  ---:  |  ---:  |
> | SSF [2] | -28.94% | -23.74% | - | - | - | - |
> | ELF-VC [6] | -3.58% | 2.03% | - | - | - | - |
> | B-EPIC [7] | -1.23% | -9.27% | -23.32% | -12.57% | -31.39% | 15.26% |
>
> >**Q10**: I wonder why the authors hide the meaningful classes B and E of the HEVC test datasets in the supplementary material.
>
> **A10**: The coding speed of VTM on high-resolution datasets (classes B and E) is very slow. We only had time to put these results in the supplementary material. We will put the results of classes B or E on the revised version of the main paper.
>
> >**Q11**: Finally, it would be great to provide more BD-rate savings (for all considered datasets and also for the MSE models).
>
> **A11**:  In the following table, we provide the BD-rate savings of VLVC against the learning-based codecs in terms of MS-SSIM.
>
> |  Codec  | UVG | MCL-JCV | ClassB | ClassC | ClassD | ClassE |
> |  :---   |  ---:  |  ---:  |  ---:  |  ---:  |  ---:  |  ---:  |
> | SSF [2] | -28.94% | -23.74% | - | - | - | - |
> | DVC_pro [3] | -51.03% | - | -47.58% | -45.16% | -50.25% | -31.99% |
> | NVC [4] | -31.34% | - | -36.59% | -42.86% | -45.86% | -24.66% |
> | RLVC [5] | -29.12% | -32.35% | - | - | - | - |
>
> In the following table, we provide the BD-rate savings of VLVC against the learning-based codecs in terms of PSNR.
>
> |  Codec  | UVG | MCL-JCV | ClassB | ClassC | ClassD | ClassE |
> |  :---   |  ---:  |  ---:  |  ---:  |  ---:  |  ---:  |  ---:  |
> | SSF [2] | -31.27% | -20.29% | - | - | - | - |
> | DVC_pro [3] | -24.16% | - | -34.49% | -14.30% | -21.64% | -3.94% |
> | NVC [4] | -31.93% | - | -34.97% | -23.84% | -32.15% | -3.59% |
> | RLVC [5] | -23.89% | -24.11% | - | - | - | - |
>
> We will try to add these statistics more or less in our revision.
>
> >**Q12**: The performance of training solely on different prediction modes.
>
> **A12**: VLVC optimized for only one prediction mode would perform slightly better than the common VLVC (about 0.1 dB) that are optimized for all prediction modes.
>
> Reference:
> - [1] Variational image compression with a scale hyperprior, ICLR 2018.
> - [2] Scale-space flow for end-to-end optimized video compression, CVPR 2020.
> - [3] An End-to-End Learning Framework for Video Compression, TPAMI 2020.
> - [4] Neural video coding using multiscale motion compensation and spatiotemporal context model, TSCVT 2020.
> - [5] Learning for Video Compression with Recurrent Auto-Encoder and Recurrent Probability Model, JSTSP 2021.
> - [6] ELF-VC: Efficient Learned Flexible-Rate Video Coding, arXiv:2104.14335.
> - [7] Extending Neural P-frame Codecs for B-frame Coding, arXiv:2104.00531.

---

> > ### Comment · Reviewer_hMDd · 2021-08-20
> > **Reply**
> >
> > Thanks to the authors for a very detailed response and several new results. The answers did clarify some aspects of the paper and the new experiments further underline how good the empirical performance of this method is.
> >
> > I now believe that a revised version of this paper that includes these new results and a better and extended discussion could be a good conference paper. However, that would represent a major change from the initial submission and requires a bit of advance credit. All in all, I am slightly more favorable towards acceptance, though it remains a borderline case for me. I have increased my rating to 6 to reflect this.

---

> > > ### Author Response · Authors · 2021-08-21
> > > **Authors' response**
> > >
> > > We appreciate your increased rating and constructive comments. We will carefully revise our paper as suggested.

---

### Official Review · Reviewer_GPmx · 2021-07-15

**Rating:** 5
**Confidence:** 5

**Summary:**

This paper introduces multi-frame references for learned video coding, which estimates multiple optical flows as voxel flows for motion compensation. With the weighted map, it can achieve weighted trilinear warping to combine the multiple predictions into  the final prediction. It is a good idea to obtain a better predcition to search information across multiple frames.  The author introduces  the polynominal motion modeling to use multiple backward optical flow to generate the single optical flow. Based on the method of softmax spaltting, they convert the backward flow into a forward flow using the flow reversal layer. Then use the forward flow and softmax spaltting for motion compensation. The overall architecture is a hybrid video coding system and the main contribution is inter coding. Residual coding is applied on the feature domain.

**Limitations And Societal Impact:**

Please see the comments below.

**Main Review:**

1) Voxel flows. The concept of voxel flow is proposed in the paper [1] for weighted prediction which can utilize the different information from different frames partially for a final prediction. The author uses this idea for the multi-frame reference for video coding with a rate-distortion optimization. Although decoding multiple flows sometimes only add some consumption of bit rates, can the author give some data about the comparison between single flow and voxel flow. Meanwhile, the author should include the MLVC [2] for comparison because this paper also uses multiple frame as references.
2) Forward flow with softmax spaltting. We think the author is aimed to use the forward flow with softmax spaltting for compensation to solve the problem of the worse predcition around the motion edges and occlusions in some extent. However, voxel flows can also solve this problem with the long-term prediction with the weighted maps. Can the author give some results when only using the generated backward flow for compensation?
3) We think the main contributions are voxel flows and the different wapring method based the proposed polynominal motion modeling. They are mainly from the existing published papers and just apply these methods to the learned video coding from the frame interpolation and prediction. The overall framework and concept are still follow the hybrid video coding system.
4) Experiments only show the results on UVG and ClassC and ClassD. How about the results on ClassE and ClassB and MCL-JCV?
5) More learned video coding methods should be added for comparison such as DVC_Pro [3], [4],[5], [6].
6) We also find that the bit rate on ClassC and ClassD are larger than 0.1 bpp. As a video coder, performance between 0.01 ~ 0.1 bpp is significant. Please provide the results between the bit rate range and include the results of the listed learned video coder.




[1]. Ziwei Liu, Raymond A Yeh, Xiaoou Tang, Yiming Liu, and Aseem Agarwala. Video frame synthesis 371
using deep voxel flow. In Proceedings of the IEEE International Conference on Computer Vision , pages 372
4463–4471, 2017.
[2]. Jianping Lin, Dong Liu, Houqiang Li, Feng Wu. M-LVC: Multiple Frames Prediction for Learned Video Compression. Proceedings of the IEEE/CVF Conference on Computer Vision and Pattern Recognition, 2020.
[3]. Guo Lu; Xiaoyun Zhang; Wanli Ouyang; Li Chen; Zhiyong Gao; Dong Xu. An End-to-End Learning Framework for Video Compression. IEEE Transactions on Pattern Analysis and Machine Intelligence 2020.
[4]. Haojie Liu, Ming Lu, Zhan Ma, Fan Wang, Zhihuang Xie, Xun Cao, Yao Wang. Neural video coding using multiscale motion compensation and spatiotemporal context model. IEEE Transactions on Circuits and Systems for Video Technology 2021.
[5]. Ren Yang, Fabian Mentzer, Luc Van Gool, Radu Timofte. Learning for Video Compression with Recurrent Auto-Encoder and Recurrent Probability Model. JSTSP 2021.
[6]. Eirikur Agustsson, David Minnen, Nick Johnston, Johannes Balle, Sung Jin Hwang, George Toderici;Scale-Space Flow for End-to-End Optimized Video Compression.  Proceedings of the IEEE/CVF Conference on Computer Vision and Pattern Recognition (CVPR), 2020, pp. 8503-8512

**Time Spent Reviewing:**

60

---

> ### Author Response · Authors · 2021-08-10
> **Authors' response**
>
> We appreciate your positive comment on our idea as well the detailed suggestions. Your concerns are addressed below.
>
> >**Q1**: Although decoding multiple flows sometimes only adds some consumption of bit rates, can the author give some data about the comparison between single flow and voxel flow.
>
> **A1**: The comparison results between single voxel flow and multiple voxel flows are shown in Figure 5a, 5b in Section 4.3. We find that motion compensation with multiple voxel flows achieves about 1dB performance gain with only a slight increase of the bitrate of flows. Thanks for your question, we will further highlight the corresponding part in our revision.
>
> >**Q2**: Meanwhile, the author should include the MLVC [1] for comparison because this paper also uses multiple frames as references.
>
> **A2**: The comparisons with MLVC [1] are provided in Figure 4 of our main paper, where MLVC is noted as Li (CVPR'20). As we can see, our method outperforms MLVC obviously. Specifically, our method achieves -33.2% BD-rate savings on the UVG dataset in terms of MS-SSIM.
>
> >**Q3**: We think the author is aimed to use the forward flow with softmax splatting for compensation to solve the problem of the worse prediction around the motion edges and occlusions in some extent. However, voxel flows can also solve this problem with the long-term prediction with the weighted maps. Can the author give some results when only using the generated backward flow for compensation?
>
> **A3**: Thanks for your comments, but it seems that you have some misunderstanding about our approach. In our method, the forward optical flow with softmax splatting is NOT used for motion compensation but for improving the compression and generation of voxel flows. As illustrated in Figure 2 of our main paper, the forward flow is directly fed into the motion decoder as conditional/assistive information to promote the generation and compression of voxel flows. Thus, in our proposed framework, we only use voxel flows (rather than optical flows) as the motion descriptors for motion compensation. The forward/backward optical flows are taken as auxiliary information that is directly concatenated into the voxel flow decoder. Thus, it makes no sense to report the results of using backward optical flow for compensation.
>
> >**Q4**: We think the main contributions are voxel flows and the different warping method based the proposed polynomial motion modeling. They are mainly from the existing published papers and just apply these methods to the learned video coding from the frame interpolation and prediction. The overall framework and concept are still follow the hybrid video coding system.
>
> **A4**: First, the core of our work is the idea of a unified/versatile neural video compression framework that is applicable to various prediction modes. This is an important demand in the field of compression which is not considered in previous works. Second, technically, although voxel flow is a tool proposed by early work, how to utilizing this tool for video compression is still challenging. Intuitively, it seems that weighted voxel flows would cost more bits for transmission compared with conventional optical flow. But we found the extra cost deserves since the transmission cost of residual would be reduced obviously. Thus, our work is not a trivial application of voxel flow but a delicate design to exploit voxel flows for the challenges in the field of compression. Third, the other contribution in this paper, i.e., our proposed polynomial motion modeling, is fully novel. The idea itself and our implementation of this idea are new and effective. It assists the compression and generation of voxel flows, obviously.
>
> >**Q5**: Experiments only show the results on UVG and ClassC and ClassD. How about the results on ClassE and ClassB and MCL-JCV? More learned video coding methods should be added for comparison such as DVC_Pro [3], [4], [5], [2].
>
> **A5**: We have provided the results on ClassE and ClassB in Figure 1 of the supplementary material. In the following table, we also provide the BD-rate savings of VLVC against the learning-based codecs [3], [4], [5], [2] in terms of MS-SSIM. Our method achieves obviously better performance compared with these works in multiple benchmark datasets. We will add these statistics in our revision.
>
> |  Codec  | UVG | MCL-JCV | ClassB | ClassC | ClassD | ClassE |
> |  :---   |  ---:  |  ---:  |  ---:  |  ---:  |  ---:  |  ---:  |
> | SSF [2] | -28.94% | -23.74% | - | - | - | - |
> | DVC_pro [3] | -51.03% | - | -47.58% | -45.16% | -50.25% | -31.99% |
> | NVC [4] | -31.34% | - | -36.59% | -42.86% | -45.86% | -24.66% |
> | RLVC [5] | -29.12% | -32.35% | - | - | - | - |
>
>
> >**Q6**: We also find that the bit rate on ClassC and ClassD are larger than 0.1 bpp. As a video coder, performance between 0.01 ~ 0.1 bpp is significant. Please provide the results between the bit rate range.
>
> **A6**: The corresponding RD points (bpp/ms-ssim) on ClassC and ClassD are 0.0695/0.9576 and 0.0611/0.9615, respectively, which are much higher than DVC_pro and NVC. These statistics are missing in our main paper due to the deadline time limit. We will add it to our revision.
>
>
> Reference:
> - [1] M-LVC: Multiple Frames Prediction for Learned Video Compression, CVPR 2020.
> - [2] Scale-space flow for end-to-end optimized video compression, CVPR 2020.
> - [3] An End-to-End Learning Framework for Video Compression, TPAMI 2020.
> - [4] Neural video coding using multiscale motion compensation and spatiotemporal context model, TSCVT 2020.
> - [5] Learning for Video Compression with Recurrent Auto-Encoder and Recurrent Probability Model, JSTSP 2021.

---

> > ### Comment · Reviewer_GPmx · 2021-08-19
> > **Reponse to the author**
> >
> > Thanks for your reply. The polynomial motion modeling is interesting and effective as reported. It is appreciated that the author can make it more clear for the readers.
> > We plan to modify the rating to 5 and leave the final decision to AC.

---

> > > ### Author Response · Authors · 2021-08-21
> > > **Authors' response**
> > >
> > > Thanks much for your affirmation on our proposed method. We made a lot of efforts in addressing a crucial problem for neural video compression, i.e., supporting various prediction modes as well as achieving SOTA rate-distortion performance with comparable complexity relative to other neural video codecs. In addition to our proposed polynomial motion modeling, the weighted voxel flow used here is effective and free for motion compensation, which would be a competitive option for future work in this field. We are looking forward to know whether you have additional concerns or suggestions for our work. If possible, we hope to try our best to get your further endorsement via more detailed technical communication. We appreciate your time and great efforts in reviewing.

---

### Official Review · Reviewer_htG5 · 2021-07-16

**Rating:** 7
**Confidence:** 4

**Summary:**

Learned video compression is a promising field and has demonstrated that it can be as effective as the classic video compression method. In terms of rate distortion it has caught up with the classical methods greatly. However binding the prediction mode and fixed network is what hinders the field to go forward. This paper proposes a versatile learned video compression framework that learns one mode to predict all possible modes. The paper proposes versatile compression where the motion compensation applies 3D motion vector fields, trilinear warping. Motion estimation here acts as a decoupler of prediction modes away from framework design, allowing its dependence to be not affected by framework design. For multiple reference frame prediction, the network predicts the motion fields with a unified polynomial function. The flow prediction can lead to reduction of computation by reducing voxel flow transmission.

**Ethical Concerns:**

See above


**Ethics Review Area:**

["I don’t know"]

**Limitations And Societal Impact:**

For video compression I cannot imagine a suitable case for ethical impact. However, any piece of technology can contribute to wrong use and ethical implications. This work did not address such concern to my knowledge.


**Main Review:**

The strengths of this paper is that it explains the motivation behind the proposed method in an understandable way and the articulation is sufficient in that it is easy to understand. The introduction well describes the intuition and motivation behind designing a versatile motion compensation module that can handle multiple prediction mode scenarios. It also explains about the polynomial motion trajectory.
Overall, the paper is well written and easy to follow with good figures explaining the concept. The experimental results also show that the proposed method performs well against other SOTA methods.
A concern is that this paper does not show any analysis about computational complexity in the main paper. The greatest reason why learning approaches are not being used widely compared to classical methods is due to computation complexity. I would suspect that incorporating a versatile 3D voxel flow module to handle all prediction mode cases will give a great impact on computational speed. Also, solving for the polynomial coefficients for the motion trajectories does not sound computationally efficient. The analysis on computational efficiency was lacking in my opinion. There needs to be more analysis on this aspect. The computational impact will be different for each prediction mode for the classical method, however, the computational load will more or less be similar for the proposed methods since it uses a unified framework for predicting the trajectories for all modes. Also more analysis is needed on the computational load for polynomial coefficient prediction.
It would be wise to compare the computational load with the classical methods as well for different prediction modes.
Given that the model size is 103MB, about 3 times as large as PWC-Net, the model seems sizable and may most likely translate to high computational complexity. Nonetheless, I believe a comparison with other Learning based methods at least is needed.


**Time Spent Reviewing:**

6

---

> ### Author Response · Authors · 2021-08-10
> **Authors' response**
>
> Thanks for your comments.
>
> >**Q1**: The analysis on computational efficiency.
>
> **A1**: Thanks for your suggestion. We will add more detailed analysis on the computational complexity in various prediction modes to our revision. Generally, the prediction modes fall into three categories: single reference frame (low delay P), unidirectional prediction with multiple reference frames (low delay B) and bidirectional prediction with multiple reference frames (random access).
> In low delay P setting, the trajectory prediction module is turned off. The difference of computational complexity between our method and previous neural video codecs mainly comes from the proposed voxel flows, which in fact has little impact on the entire complexity. Our proposed weighted voxel flow based warping takes about 0.033s per frame for HD 1080. In the multiple-reference-prediction scenario, the complexity of our model is increased similar to other reference coding methods. The complexities under unidirectional and bidirectional predictions with multiple flows are similar since we apply a unified multiple-reference prediction module. The complexities of our model in these scenarios are in fact comparable with previous works which also adopt multiple reference frames. We here provide some statistical results on the network inference time of our method and other methods, such as DVC [1] (single reference frame) and MLVC [2] (multiple reference frames).
>
> |    | DVC | VLVC (LDP) | MLVC | VLVC (LDB) | VLVC (RA) |
> |  :---:   |  :---:  |  :---:  |  :---:  |  :---:  |  :---:  |
> | encoding time (s) | 0.59 | 0.46 | 1.23 | 0.86 | 0.86 |
> | decoding time (s) | 0.32 | 0.33 | 0.99 | 0.74 | 0.74 |
>
>
> The runtime of our method is slightly less than DVC and MLVC under similar coding configurations.  For a fair comparison,  we don’t include the time of arithmetic coding because it is a common part of any codec and is sensitive to implementation. For VLVC, the overall coding speed under the random access mode is about 0.7 fps.
>
> In addition, about the model size, our model size is numerically large (about 70Mb except for PWC-Net) since we apply three residual blocks after each downsampling/upsampling layer to enhance the motion/residual compression network, compared with DVC (about 11MB). This is a trivial enhancement that leaves a large room for model slimming. We just try to build a model with consistent designs.
>
> Reference:
> - [1] An End-to-End Learning Framework for Video Compression, TPAMI 2020.
> - [2] M-LVC: Multiple Frames Prediction for Learned Video Compression, CVPR 2020.

---

> > ### Comment · Reviewer_htG5 · 2021-08-24
> > **Reply**
> >
> > I have read the other reviews and the author responses. I am willing to maintain my initial rating of accept since I still believe the paper to be a good addition to the community.

---

> > > ### Author Response · Authors · 2021-08-29
> > > **Authors' response**
> > >
> > > Thank you very much for your affirmation and constructive comments.

---

### Official Review · Reviewer_BSR1 · 2021-07-19

**Rating:** 6
**Confidence:** 4

**Summary:**

This paper proposes a neural video compression method that is designed around the idea of using a voxel flow instead of the more traditional pixel warping idea. The key difference here is that a volume of reference frames is used, and flow is computed to all of them. This allows the compression method to be quite liberal in terms of what information from the GOP can be used, yielding a better RD performance than methods that rely on a single reference frame.

Overall I think this is a step in the right direction, but I fear that such a method is completely impractical for the foreseeable future.

**Limitations And Societal Impact:**

The limitations are really not discussed at all. This method has some glaring problems I mentioned in the review and they're not at all acknowledged.

**Main Review:**

This method makes use of a frame volume that can be equal to the number of frames in the GOP. To give some perspective, normal video codecs tend to be very memory sensitive, and in general it's not possible to have more than a small set of reference frames due to the huge memory pressure that this imposes, and of course cost (more frames = more RAM needed). This problem is exacerbated by the fact that we see a trend towards higher resolutions which grow in size quadratically, making the cost of multiple reference frames completely prohibitive. This is my main worry about the applicability of this paper... I am not sure how realistic the assumption is that a copy of N reference frames can be kept in RAM at decoding time....where N >> 2. Similarly the number of flows is also on the order N, making my worry even worse.

Ignoring the practical aspect, the paper presents an interesting approach to improve compression. I haven't seen something done exactly this way in any other area so I think that we have an increased level of originality here. The main differentiator is the multiple flow volume idea (multiple reference frames have been done many times).

I applaud the authors for not fearing to compare against HM, because that's the best possible case for HEVC. I am not familiar enough with the settings to comment whether the modifications made to the profile used make sense, and I would invite a reviewer with more familiarity to comment on this.

I also think it's good that the details of the network are presented in the supplementary materials.

My second biggest concern, though is related to runtime. This method sounds very very expensive and at least in the body of the main paper this issue is not really addressed. I think in the final version THIS MUST be explicitly addressed.

Overall, I am inclined to accept this paper.

**Time Spent Reviewing:**

3

---

> ### Author Response · Authors · 2021-08-10
> **Authors' response**
>
> Thanks for your positive comments on the right direction of our work and valuable suggestions. Overall, learning-based coding is a critical technological innovation that is taking place. We believe it will become more and more practical as the computation power develops. We try to address your concerns as below.
>
> >**Q1**: This method makes use of a frame volume that can be equal to the number of frames in the GOP. To give some perspective, normal video codecs tend to be very memory sensitive, and in general it's not possible to have more than a small set of reference frames due to the huge memory pressure that this imposes, and of course cost (more frames = more RAM needed). How realistic the assumption is that a copy of N reference frames can be kept in RAM at decoding time....where N >> 2?
>
> **A1**: We agree that the memory cost is an important factor to be considered in developing codecs, not only for learning-based codecs but also for conventional codecs. However, we would like to point out that the memory cost is not our limitation due to the following aspects: 1) The memory cost of reference frames is in fact a common concern for both conventional and deep-learning-based codec. Our proposed framework is comparable with conventional video coding standard VVC that supports up to 8 reference frames. In practice, we adopt 3 reference frames in our proposed VLVC by default and use up to 6 reference frames in our experiments, although there is no limitation for the number of used reference frames. 2) The memory cost of our proposed framework is also scalable to the number of voxel flows. As shown in Figure 5(a), with our proposed design, using 4 flows achieve comparable performance compared to using 25 flows. This indicates that we can significantly improve the efficiency with a solely slight performance drop. Our proposed framework can support flexibly adjusting the number of voxel flows towards more efficient deployment for practical applications. Besides, we can relieve the memory pressure via implementation optimization, such as serial flow-based warping.
>
> >**Q2**: My second biggest concern, though is related to runtime. This method sounds very very expensive and at least in the body of the main paper this issue is not really addressed. I think in the final version THIS MUST be explicitly addressed.
>
> **A2**: Thanks for your suggestion. We will move our description about the runtime from our supplementary to the main paper in our revised paper. The overall runtime of VLVC is comparable with recent learning-based codecs, such as DVC [1] (single reference frame) and MLVC [2] (multiple reference frames). For a fair comparison, we reimplement the works of DVC and MLVC using PyTorch and compare the network inference time on 1080p videos, except for the time of arithmetic coding (on CPU). As shown in the table, the VLVC (LDP), VLVC (LDB) and VLVC (RA) are low delay P (unidirectional, single reference frame), low delay B (unidirectional, multiple reference frames) and random access (bidirectional, multiple reference frames) modes of VLVC, respectively. The runtime of our method is slightly less than DVC and MLVC under similar coding configurations. The time of arithmetic coding is not included because it is a common part of any codec and is sensitive to implementation. For VLVC, the overall coding speed of RA mode is about 0.7 fps.
>
> |    | DVC | VLVC (LDP) | MLVC | VLVC (LDB) | VLVC (RA) |
> |  :---:   |  :---:  |  :---:  |  :---:  |  :---:  |  :---:  |
> | encoding time (s) | 0.59 | 0.46 | 1.23 | 0.86 | 0.86 |
> | decoding time (s) | 0.32 | 0.33 | 0.99 | 0.74 | 0.74 |
>
> Reference:
> - [1] An End-to-End Learning Framework for Video Compression, TPAMI 2020.
> - [2] M-LVC: Multiple Frames Prediction for Learned Video Compression, CVPR 2020.

---

> > ### Comment · Reviewer_BSR1 · 2021-08-17
> > **reply**
> >
> > Thank you for taking your time to do additional measurements.
> >
> > With respect to not including the entropy coding time, I think it's a bit misleading to not include it. When you do use VVC for example, the entropy coder they use is definitely used (both for encoding/decoding). If we are to have real-world numbers showcased to the world, it's quite important to be clear about the runtime. With that said, I still think it's an improvement to have these numbers vs. not having them. In any case, given that you're essentially comparing the neural aspect for all methods, I am ok with this.
> >
> > I can be convinced by the others/AC that we should accept this.

---

> > > ### Author Response · Authors · 2021-08-20
> > > **Response**
> > >
> > > Thank you for your reply.
> > >
> > > The entropy coding is a common part for any codec and is sensitive to implementations. During test, the implementation of this part is commonly off-the-shelf where different compression models can use the same one. Thus, we exclude this part for fairer comparison on the model complexity with other neural video codecs. Besides, please note that the encoding/decoding time (about 1.5 sec per frame for HD 1080 decoding) reported in the supplementary material has included the time of entropy coding part. Here, we apply the off-the-shelf range-coder package without any optimization, which can be further accelerated in engineering.

---

### Author Response · Authors · 2021-08-20
**Feedback Request**

We thank all the reviewers for the thoughtful review comments and constructive suggestions. We would like to know if our response has addressed your concerns. We tried our best to address the main concerns raised in your first review, including, 1) the coding complexity of different prediction modes, 2) the detailed explanation for polynomial motion modeling and 3) more comparisons on the R-D performance with other methods. In short, our method can support various prediction modes with state-of-the-art R-D performance and comparable time complexity, compared to existing learning-based codecs. We believe that weighted voxel flows will be a competitive option for future work in this field. Our proposed polynomial motion modeling is also effective for multiple reference prediction. Please feel free to leave your comments if there are any further questions or suggestions.

---

### Decision · Program_Chairs · 2021-09-27

**Decision:**

Reject

**Comment:**

The work proposes and implements a learning-based video compression solution, which is able to handle multiple prediction modes and achieves good rate-distortion performance. The key building blocks are not new, but the actual model and its successful implementation bring an interesting contribution to the video compression problem.
The submitted manuscript can be completed by the multiple additional results discussed with the reviewers, and a complexity analysis; the presentation should finally be improved before an eventual publication. In particular, the work should be more clearly positioned wrt to recent competitors, including 'T. Ladune et al, Conditional coding for flexible learned video compression, ICLR 2021 workshop neural compression'.
While the work is really interesting, the revisions that result from the active discussion with the reviewers will lead to a quite different, improved, manuscript, which probably deserves another review before a positive outcome.